# MORPH: PDE FOUNDATION MODELS WITH ARBITRARY DATA MODALITY

## ABSTRACT

We introduce MORPH, a modality-agnostic, autoregressive foundation model for partial differential equations (PDEs). MORPH is built on a convolutional vision transformer backbone that seamlessly handles heterogeneous spatiotemporal datasets of varying data modality (1D–3D) at different resolutions, and multiple fields with mixed scalar and vector components. The architecture combines (i) component-wise convolution, which jointly processes scalar and vector channels to capture local interactions, (ii) inter-field cross-attention, which models and selectively propagates information between different physical fields, (iii) axial attentions, which factorize full spatiotemporal self-attention along individual spatial and temporal axes to reduce computational burden while retaining expressivity. We pretrain multiple model variants on a diverse collection of heterogeneous PDE datasets and evaluate transfer to a range of downstream prediction tasks. Using both full-model fine-tuning and parameter-efficient low-rank adapters (LoRA), MORPH outperforms models trained from scratch. Across extensive evaluations, MORPH matches or surpasses strong baselines and recent state-of-the-art models. Collectively, these capabilities present a flexible and powerful backbone for learning from the heterogeneous and multimodal nature of scientific observations, charting a path toward scalable and data-efficient scientific machine learning.

## 1 INTRODUCTION

Many problems in physics are governed by partial differential equations (PDEs) where field variables exhibit complex spatiotemporal evolution. Machine learning-based surrogate models have made substantial progress in learning such spatiotemporal physical systems. Prominent standalone surrogates include neural operators such as DeepONet (Lu et al., 2021; Kontolati et al., 2024), Fourier Neural Operators (Li et al., 2020), the U-shaped Neural Operator (Rahman et al., 2022) and transformer-based models (Solera-Rico et al., 2024; Li et al., 2022). Inspired by the success of foundation models in natural language processing, recent research directions in scientific machine learning are transitioning from task-specific to task-agnostic modeling. We use the term "PDE foundation model" to denote a universal PDE surrogate or a universal PDE operator pretrained on large, diverse datasets spanning multiple physics. With task-specific fine-tuning, such models can outperform standalone surrogates and demonstrate utility in data- and compute-scarce scenarios.

*Partial observability of physics is a bottleneck in learning a PDE foundation model.* Two forces collide in practice. *Physics:* dynamical systems couple nonlinear, multiscale interactions across scalar and vector fields evolving in four-dimensional continuous space-time, often with widely separated characteristic scales. *Data:* unlike web-scale text or images, pretraining corpora for PDEs are scarce, costly to curate, and frequently provide only partial observability. The existing benchmark spatiotemporal PDE datasets (Takamoto et al., 2022; Ohana et al., 2024; Herde et al., 2024) contain varying data modality (1D–3D) including trajectories, fields, components and resolutions curated through computationally costly simulations. The datasets are generated under restricted parameter settings with limited variation in initial and boundary conditions. In addition, such datasets can be terabytes in size, making naive homogenization or padding prohibitive. On the experimental side, measuring such fields demands substantial investment and dedicated facilities (e.g., wind tunnels for fluid-dynamics experiments). Most measurement modalities also provide partial observability: seismometers, for instance, yield 1D time series on sparse spatial grids, whereas seismic waves propagate as a $(3+1)$-D disturbance through Earth's crust. Similarly, pressure probes and ve-

locimetry typically provide 1D or 2D temporal observations at limited spatial locations, even though the underlying fluid physics exhibits full spatiotemporal fluctuations in the medium. These practical constraints of simulations and measurements motivate a design that can *learn effectively from partial information* and *generalize across data heterogeneity*.

*A modeling challenge under partial observations.* A foundation model with a transformer backbone provides flexibility in learning from partial physical information. Recent PDE foundation models have made substantial progress toward a task-agnostic universal surrogate. However, most of the works (McCabe et al., 2024; Chen et al., 2024; Herde et al., 2024) assume largely homogeneous data modality, with predominantly 2D Cartesian grids, fixed scalar or vector components of field variables, and limited multiphysics coverage. The architectural choices are not designed for data heterogeneity and scale unfavorably as data modality changes and resolution increases often causing memory/compute blow-ups while requiring task-specific reconfiguration during fine-tuning. In particular, their pretraining sets exclude 3D datasets because patching volumetric data expands the token sequence length, driving quadratic self-attention cost $\mathcal{O}(L_{\text{seq}}^2)$. The same scaling bottleneck discourages inclusion of higher-resolution 2D datasets. Conversely, 1D problems are often excluded because 2D-centric pipelines enforce padding or tiling up to a 2D layout, wasting memory and compute without adding information. Since many experimental measurements are practically 1D (e.g., point probes, line scans, seismometer traces) and provide the most accurate observation of the underlying physics, we regard 1D as a crucial modality. We therefore treat heterogeneity and multi-modality as first-class design requirements for a PDE foundation model. Concurrent works like (Ye et al., 2024; McCabe et al., 2024) explicitly highlight the open challenge of supporting more complex domains and extending beyond 2D, underscoring a persistent heterogeneity gap. We identify this as a key bottleneck in current PDE foundation model research. In short, there is a growing need for a flexible and scalable foundation model that can learn from diverse forms of partially observed scientific data.

**Our Approach.** We introduce MORPH, a PDE foundation model designed to accommodate heterogeneous data across diverse physical phenomena. MORPH is shape-agnostic, with physics-aware channel handling of PDE datasets. The architecture combines three mechanisms: (a) component-wise convolutions to capture local interactions while jointly processing scalar and vector channels; (b) inter-field cross-attention that models and selectively propagates information across physical fields while condensing them into a single fused representation; and (c) axial attention that factorizes full spatiotemporal self-attention along spatial and temporal axes, reducing compute while retaining expressivity. We define a Unified Physics Tensor Format (UPTF-7) for mini-batches that generalizes across PDE datasets while preserving the semantics of the physical observations. We pretrain MORPH on six heterogeneous spatiotemporal datasets spanning 1D–3D spatial domains with multiple field compositions. We perform full finetuning and parameter-efficient finetuning with low rank adapters (LoRA) on seven additional heterogeneous datasets for downstream prediction tasks. We release four model variants: MORPH-Ti (7M), MORPH-S (30M), MORPH-M (126M), and MORPH-L (480M).

Our specific contributions are:

- **Modality-agnostic universality.** We introduce a modality-agnostic model that operates across arbitrary spatiotemporal PDE dataset dimensionalities (1D–3D), resolutions and mixed scalar/vector fields without task-specific reconfiguration.

- **Expanded spatiotemporal context.** MORPH employs a larger transformer architecture with one cross-attention and four axial attention modules that attend over a *multi-fold* increase in spatiotemporal patches (i.e., a larger context window).

- **Diverse pretraining and transfer.** We pretrain and fine-tune on a broad (3 benchmarks) heterogeneous suite including multi-physics datasets like magnetohydrodynamics (MHD), turbulent self-gravitating flows with cooling (TGC), high-resolution 2D compressible and incompressible Navier–Stokes and large-scale 3D datasets.

- **Parameter-efficient adaptation.** We adapt LoRA for fine-tuning our largest model (MORPH-L), showing that low-rank adapters recover most of the gains of full fine-tuning with substantially fewer trainable parameters.

## 2 RELATED WORK

**Standalone PDE surrogate models.** A wide range of ML-based standalone surrogate models are deployed for solving PDEs. Some of these methods are 3D-CNN (Wandel et al., 2021), ConvLSTM (Shi et al., 2015), DCGAN (Cheng et al., 2020) and GNNs (Chen et al., 2021). Beyond purely data-driven surrogates, physics-informed neural networks (PINNs) have been applied to spatiotemporal problems (Raissi et al., 2019). Recently, neural operators like DeepONet (Goswami et al., 2022), Fourier Neural Operators (FNOs) (Li et al., 2020), wavelet neural operators (WNOs) (Tripura & Chakraborty, 2023), Laplace neural operators (Cao et al., 2024) and physics-informed neural operators (PINOs) (Li et al., 2024b) have gained attention for solving spatiotemporal problems. Another line of methods uses generative models to capture spatiotemporal data distributions. Generative Adversarial Networks (GANs) have been explored to reconstruct or super-resolve turbulent flow fields from partial observations (Buzzicotti et al., 2021). More recently, diffusion models with spatial cross-attention (score-based generative models) have gained traction due to their stability and high fidelity in representing complex distributions (Zhuang et al., 2025; Li et al., 2024a; Gao et al., 2025). Researchers have investigated latent evolution models where spatial learning is performed through autoencoders (Oommen et al., 2022; Vlachas et al., 2022; Maulik et al., 2021; Kontolati et al., 2024) and variational autoencoders (Solera-Rico et al., 2024; Rautela et al., 2024). Temporal dynamics in the latent space have been modeled with recurrent neural networks including LSTMs (Maulik et al., 2021; Vlachas et al., 2022; Liu et al., 2022; Rautela et al., 2024), Koopman with non-linear forcing (Khodkar & Hassanzadeh, 2021), DeepONets (Oommen et al., 2022; Kontolati et al., 2024), ODE-Nets (Chen et al., 2018) and transformers with self-attention and easy-attention (Solera-Rico et al., 2024; Alkin et al., 2024).

**PDE foundation models.** Standalone PDE surrogates are typically specialized to one equation or parameterized family, which means they must be retrained from scratch whenever the PDE form or conditions change. This lack of generality leads to significant computational overhead without knowledge transfer across PDEs. Some initial works have successfully demonstrated the capability of transfer learning for PDEs (Goswami et al., 2022; Xu et al., 2023). Foundation models for PDEs have emerged to overcome these limitations by capturing the common patterns underlying many different PDE systems. A PDE foundation model is a task-agnostic generalist model pretrained on diverse data, followed with minimal task-specific finetuning. Recent works (Yang et al., 2023; Chen et al., 2024; Herde et al., 2024; Chen et al., 2018; Liu et al., 2024; Ye et al., 2024) demonstrated that a PDE foundation model can outperform standalone models. The advantages in generality, data efficiency, transferability, and versatility motivate the development of foundation PDE models over traditional one-off surrogates (Subramanian et al., 2023). Despite rapid progress, building general-purpose PDE foundation models remains a formidable challenge. PDEs capture complex physics and the datasets representing such spatiotemporal behavior involve several human experts and many compute hours. Public benchmarks remain scarce and frequently provide only partial observability of the underlying physics (Takamoto et al., 2022; Ohana et al., 2024; Herde et al., 2024). It becomes absolutely necessary to build a PDE foundation model with data heterogeneity and modalities in mind (Ye et al., 2024; McCabe et al., 2024; Morel et al., 2025). The architectural design should support learning from partial experimental/simulation data and accommodate 1D–3D domains, low to high resolutions, and mixed scalar and vector fields in single and multiphysics settings. However, most of the current PDE foundation models lack this ability.

## 3 PROPOSED METHOD

### 3.1 UNIFIED PHYSICS TENSOR FORMAT (UPTF-7)

The most direct way to handle data heterogeneity is to pad the datasets to a common shape. However, raw PDE datasets can occupy gigabytes to terabytes of storage. For instance, THE WELL alone contains $\sim$15 TB of physics simulations (Ohana et al., 2024). This makes naive padding both compute and storage-wise inefficient. We therefore seek a unified dataset format that generalizes across PDE datasets while preserving the semantics of the physical observations. We observe that popular benchmarks like PDEBENCH (Takamoto et al., 2022), THE WELL (Ohana et al., 2024), PDEGYM (Herde et al., 2024), PDEARENA (Gupta & Brandstetter, 2022), and CFD-BENCH (Yining et al., 2023) can be represented with a generic 7D data shape. We denote it as

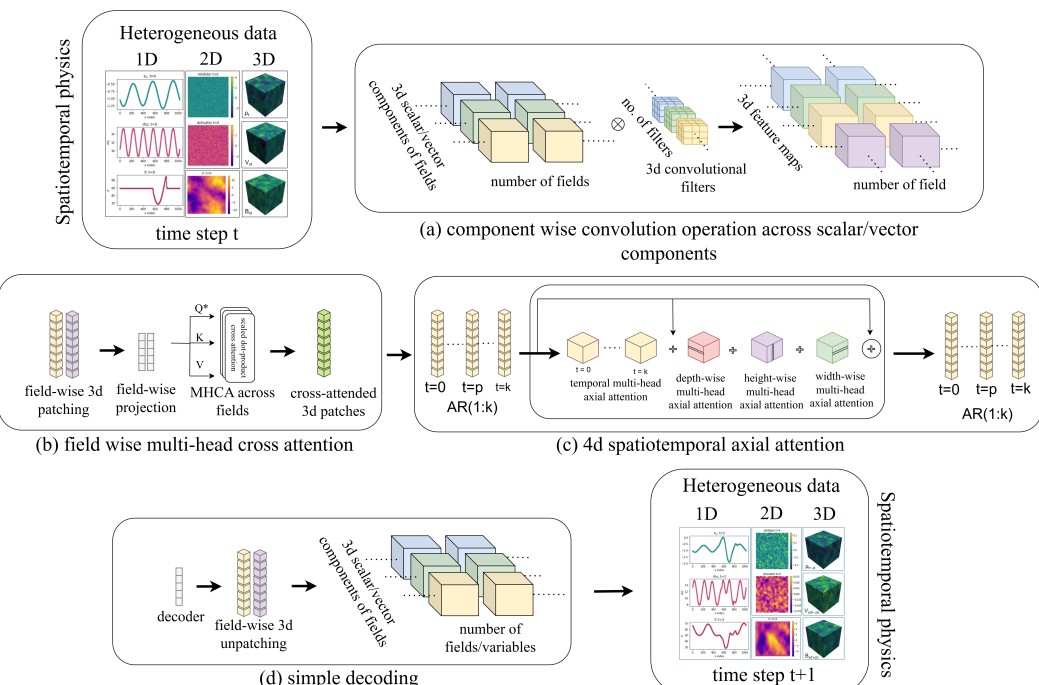

Figure 1: An illustration of the model architecture. MORPH is a shape-agnostic design that seamlessly handles heterogeneous datasets. The design consists of (a) 3D convolution operation is performed along the component ($C$) dimension providing filters× feature maps, (b) multi-head cross-attention is performed across fields ($F$) resulting in a fused field, (c) 4D factorized axial attention is performed along space-time dimension ($T, D, H, W$), (d) simple decoder maps back to the data space.

$(N, T, F, C, D, H, W)$, where $N$ denotes the number of trajectories, $T$ are the number of time steps, $N \times T$ are the number of snapshots, $F$ are the number of field variables, $C$ are the associated scalar/vector components per field, and $(D, H, W)$ are the spatial depth, height, and width. We retain each benchmark's native on-disk format and convert mini-batches on the fly to our *Unified Physics Tensor Format* (UPTF-7) when loading to GPUs. For instance, the 2D compressible CFD data in PDEBENCH contains velocity, pressure, and density fields where velocity is vector-valued while the other two are scalar. Under our convention, scalars are special cases of vectors. Therefore, we broadcast each scalar component to the maximum vector components count when a batch contains mixed scalar–vector fields. Therefore, the 2D compressible CFD batch maps to a UPTF-7 layout $(B, T, F{=}3, C{=}2, D{=}1, H, W)$. Similarly, a 1D Diffusion–Reaction batch with native $(B, T, W)$ layout maps to UPTF-7 as $(B, T, F{=}1, C{=}1, D{=}1, H{=}1, W)$, and a 3D MHD batch in THE WELL maps to $(B, T, F{=}3, C{=}3, D, H, W)$. We defer more dataset-specific details to Appendix A.4.

## 3.2 OVERVIEW OF MORPH.

A schematic of the model is shown in Fig. 1. The shape-agnostic design of MORPH handles data heterogeneity via data-channel-specific operators. The architecture comprises three key mechanisms: (a) *component-wise convolutions*, which jointly process scalar and vector channels to capture local interactions; (b) *inter-field multi-head cross-attention*, which models and selectively propagates information across physical fields; and (c) *4D axial attentions*, which factorize full spatiotemporal self-attention along individual spatial and temporal axes to reduce computational burden while retaining expressivity.

The *convolutional operator* acts on the component channel(s), producing a set of feature maps determined by the number of filters. Adding convolutional inductive bias helps vision transformers capture locality and improves sample efficiency (Dosovitskiy et al., 2020). We can scale feature

maps and ultimately MORPH via the number of convolutional filters (default: 8). We partition feature maps into patches of size 8. Across pretraining and fine-tuning datasets, we use at most 4,096 patches, yielding a 32,768-dimensional feature vector, which we linearly project to an embedding whose dimensionality ranges from 256 to 1,024 depending on the model variant.

In most PDE datasets, multiple fields with different spatial scales evolve at various temporal rates. *Field-wise multi-head cross-attention* prioritizes relevant inter-field interactions and selectively propagates information, enabling careful feature fusion while suppressing spurious activations arising from scale mismatches among coupled fields. The cross-attention operator outputs a single fused field, significantly reducing computational cost (Nguyen et al., 2023). We scale MORPH further via the cross-attention dimension, which by default matches those of axial self-attention.

We factorize full 4D spatiotemporal attention for a (3+1)D dynamical system into time-wise, depth-wise, height-wise, and width-wise *axial attentions*. For high-dimensional datasets, this limits the computational cost to $O(T^2 + D^2 + H^2 + W^2)$, in contrast to full spatiotemporal attention with cost $O((TDHW)^2)$ (Ho et al., 2019). Time-wise axial attention supports flexible temporal context allowing fixed or variable order autoregression (default: 1 step). More details on modeling aspects are deferred to Appendix B.1.

### 3.3 TRANSFER ACROSS DATA MODALITIES

A central question for a PDE foundation model is *whether representations learned on one modality (e.g., 2D incompressible flows) transfer across multiple data modalities such as 1D, 2D and 3D*. MORPH is explicitly designed to be modality-agnostic while maintaining generalization across diverse physics. To test this hypothesis, we pretrain MORPH exclusively on the 2D incompressible Navier–Stokes dataset (CFD2D-IC) under an autoregressive objective. We then perform *zero-shot inference* (no fine-tuning, all weights frozen) on six heterogeneous targets (fluids based) spanning different modalities: CFD1D, CFD2D, CFD3D, CFD3D-TURB, MHD3D, and TGC3D.

To quantify transfer, we define zero-shot *Gap-Closure Ratio* (GCR), which measures what fraction of the performance gap between a naive, training-free baseline and a fully trained-from-scratch model is closed by the pretrained model in the zero-shot setting. Formally,

$$\text{GCR} = \frac{\text{Naive Baseline Gain of the pretrained model (NBG-PT)}}{\text{Naive Baseline Gain of the trained-from-scratch model (NBG-SC)}} = \frac{1 - (E_{\text{PT}}/E_{\text{NB}})}{1 - (E_{\text{SC}}/E_{\text{NB}})} \tag{1}$$

Here, $E_{\text{PT}}$ is the zero-shot error of the CFD2D-IC pretrained model evaluated on a given target, $E_{\text{SC}}$ is the error of an identical architecture trained fully from scratch on that target, and $E_{\text{NB}}$ is the error of a training-free naive baseline (the same architecture with random initialization and no learning, informally corresponding to near–coin-toss performance), taken as the minimum over 25 random seeds. We define $1 - E_{\text{PT}}/E_{\text{NB}}$ as the naive-baseline gain of the pretrained model (NBG-PT), and $1 - E_{\text{SC}}/E_{\text{NB}}$ as the naive-baseline gain of the trained-from-scratch model (NBG-SC). In simpler terms, GCR is the ratio NBG-PT and NBG-SC. Our definition of zero-shot GCR is inspired by performance-gap-recovered (PGR) metric introduced in Burns et al. (2023).

*Interpretation*: GCR = 0 means zero-shot does no better than naive. GCR = 1 means zero-shot matches a fully trained scratch model; GCR > 1 indicates zero-shot surpasses a model trained from scratch; any GCR > 0 demonstrates positive transfer across modalities and physics. Fig. 2 summarizes transfer from CFD2D-IC to the six targets via three quantities: NBG-PT, NBG-SC and the resulting GCR. We observe consistent positive transfer (GCR > 0) across all fluid-specific targets (green bar), with the strongest effects on near-domain problems (e.g., compressible CFD1D, CFD2D and CFD3D) and attenuated but still nontrivial transfer to CFD3D (with turbulence) and multiphysics settings such as MHD3D and self-gravitating TGC3D. These results indicate that 2D pretraining learns reusable operators that carry to 1D/2D/3D data modalities, providing empirical support for MORPH's modality-agnostic design.

## 4 EXPERIMENTS

**Datasets.** We select datasets from three publicly available PDE benchmarks, i.e., PDEBENCH (Takamoto et al., 2022), PDEGYM (Herde et al., 2024) and THE WELL (Ohana et al., 2024). They

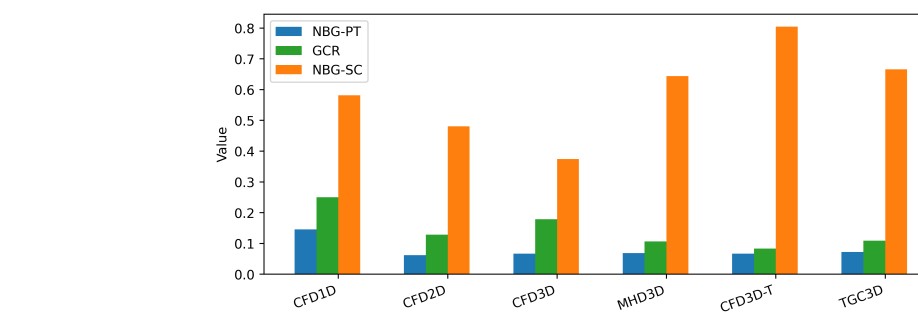

Figure 2: **Zero-shot transfer from CFD2D-IC pretraining.** Targets: CFD1D, CFD2D, CFD3D, MHD3D, CFD3D-TURB, TGC3D. Bars show the Naive Baseline Gain of the pretrained model (**NBG-PT**), the Naive Baseline Gain of a trained-from-scratch model (**NBG-SC**), and the resulting **Gap-Closure Ratio (GCR)**. GCR $> 0$ indicates cross-modality transfer.

cover a wide variety of PDEs; the latter, however, focuses on more complex multiphysics problems. Because MORPH accommodates data heterogeneity, we demonstrate versatility in our dataset selection for both pretraining and fine-tuning. The pretraining and finetuning datasets and their shapes are listed in Table 1 and 2. Compared to existing PDE foundation models, our pretraining and fine-tuning sets are larger and more diverse. We defer more dataset-specific details to Appendix A.

Table 1: Pretraining datasets with their physical fields (scalar or vector) and data shapes (N,T,D,H,W,C), together with the benchmark/source.

| PT sets | MHD-3D | DR-2D | CFD-1D | CFD2D-IC | CFD-3D | SW-2D |
|---|---|---|---|---|---|---|
| Shapes (fields) | $\rho$: $(97, 100, 64^3)$ 
 $\mathbf{B}$: $(97, 100, 64^3, 3)$ 
 $\mathbf{v}$: $(97, 100, 64^3, 3)$ | $u_1$: $(1k, 100, 128^2)$ 
 $u_2$: $(1k, 100, 128^2)$ | $\rho$: $(10k, 100, 1024)$ 
 $\mathbf{v}$: $(10k, 100, 1024)$ 
 $P$: $(10k, 100, 1024)$ | $\rho$: $(64, 1k, 512^2)$ 
 $\mathbf{v}$: $(64, 1k, 512^2, 2)$ 
 $P$: $(64, 1k, 512^2)$ | $\rho$: $(200, 21, 128^3)$ 
 $\mathbf{v}$: $(200, 21, 128^3, 3)$ 
 $P$: $(200, 21, 128^3)$ | $h$: $(1k, 100, 128^2)$ |
| Source | The Well | PDEBench | PDEBench | PDEBench | PDEBench | PDEBench |

Table 2: Finetuning datasets with their physical fields (scalar or vector) and data shapes (N,T,D,H,W,C), together with the benchmark/source.

| FT sets | DR-1D | CFD-2D | CFD3D-TURB | BE-1D | GSDR-2D | TGC-3D | FNS-KF-2D |
|---|---|---|---|---|---|---|---|
| Shapes (fields) | $u_1$: $(10k, 100, 1024)$ | $\rho$: $(10k, 21, 512^2)$ 
 $\mathbf{v}$: $(10k, 21, 512^2, 2)$ 
 $P$: $(10k, 21, 512^2)$ | $\rho$: $(600, 21, 64^3)$ 
 $\mathbf{v}$: $(600, 21, 64^3, 3)$ 
 $P$: $(600, 21, 64^3)$ | $u$: $(10k, 200, 1024)$ | $A$: $(200, 1k, 128^2)$ 
 $B$: $(200, 1k, 128^2)$ | $\rho$: $(100, 50, 64^3)$ 
 $\mathbf{v}$: $(100, 50, 64^3, 3)$ 
 $P$: $(100, 50, 64^3)$ 
 $T$: $(100, 50, 64^3)$ | $u$: $(20k, 21, 128^2)$ 
 $v$: $(20k, 21, 128^2)$ |
| Source | PDEBench | PDEBench | PDEBench | PDEBench | The Well | The Well | PDEgym |

**Pretraining and finetuning settings.** Our model accommodates both fixed–order autoregressions, $AR(p)$, and time-varying orders, $AR(p_{1:T})$, where $p$ (or $p_t$) denotes the autoregressive order. In this paper, we train MORPH in a self-supervised $AR(1)$ setting where we learn a map $F_\theta : \mathcal{X} \to \mathcal{X}$ such that for all $t \geq 0$, $X_{t+1} \approx F_\theta(X_t)$ by minimizing the mean-squared error. High–order fixed AR models (e.g., $AR(16)$ in McCabe et al. (2024) and $AR(10)$ in Hao et al. (2024)) often yield stronger long-horizon rollouts by reducing exposure bias via a longer receptive field. However, this design departs from the initial-value-problem (IVP) semantics that PDE foundation models aim to respect. For these reasons we adopt a one-step ($AR(1)$) formulation aligned with IVP semantics that preserves the compositional structure fundamental to PDE time evolution. Nevertheless, MORPH remains flexible and supports multi-order (fixed or varying) autoregression to improve rollout quality. For brevity, full pretraining and finetuning settings are documented in Appendix B.2, B.3.

## 4.1 BASELINES

**Models.** We compare MORPH against classical neural operators and recent PDE foundation models. As popular baselines, we use Fourier Neural Operator (FNO) (Li et al., 2020), a parameter-efficient spectral neural operator widely adopted for fluid surrogates, and UNet (Ronneberger et al., 2015; Rahman et al., 2022), a convolutional encoder–decoder with multiscale skip connections. Among foundation models, we include MPP (McCabe et al., 2024), which demonstrates gains from multi-physics pretraining; DPOT (Hao et al., 2024), a denoising operator transformer with Fourier

Table 3: **Standalone surrogates**: NRMSE (N) and VRMSE (V) of the experiments (lower is better) on pretraining and finetuning datasets. The *best score* is typeset in **bold**. The *global best* for the dataset is **bold + underlined**. Our standalone models are shown in blue text.

| Models / Datasets | FNO PB/TW | UNet PB/TW | MPP-SS Ti (7M) | MPP-SS S (30M) | DPOT-SS Ti (7M) | DPOT-SS S (30M) | MORPH-SS Ti (7M) | MORPH-SS S (30M) |
|---|---|---|---|---|---|---|---|---|
| Pretraining sets | | | | | | | | |
| 1D-CFD (N) | 0.095 | 0.36 | - | - | - | - | 0.06102 | **0.05719** |
| 2D-DR (N) | 0.12 | 0.84 | 0.0168 | **0.0112** | 0.0321 | 0.0379 | 0.14637 | 0.12285 |
| 2D-CFD-IC (N) | - | - | - | - | - | - | **0.08760** | 0.11385 |
| 2D-SW (N) | 0.0044 | 0.083 | 0.0066 | 0.0024 | 0.00560 | 0.00657 | 0.00445 | **0.0021** |
| 3D-MHD (V) | 0.36 | **0.1798** | - | - | - | - | 0.3149 | 0.26845 |
| 3D-CFD (N) | 0.37 | 1.0 | 0.299 | - | 0.262 | - | **0.09819** | 0.10573 |
| Finetuning sets | | | | | | | | |
| 1D-DR (N) | 0.0014 | 0.006 | - | - | - | - | 0.00158 | **0.00134** |
| 1D-BE (N) | **0.029** | 0.37 | - | - | - | - | 0.03961 | 0.07646 |
| 2D-CFD (N) | 0.28 | 0.66 | 0.0312 | 0.0213 | 0.0176 | **0.0153** | 0.05318 | 0.05712 |
| 2D-GSDR (V) | 0.1365 | 0.2252 | - | - | - | - | **0.0086** | 0.01151 |
| 3D-CFD-Turb (N) | 0.24 | 0.23 | 0.098 | - | - | - | 0.08297 | **0.07941** |
| 3D-TGC (V) | 0.2429 | 0.6753 | - | - | - | - | 0.16366 | **0.07153** |

Table 4: **Zero-shot performance of foundation models**: NRMSE (N) and VRMSE (V) of the experiments (lower is better). The *best score* is typeset in **bold**. The *global best* for the dataset is **bold + underlined**. Our foundation models in red.

| Zero-shot on pretraining sets | | | | | Zero-shot on finetuning sets | | | | |
|---|---|---|---|---|---|---|---|---|---|
| Models / Datasets | Ti (7M) | S (30M) | M (126M) | L (480M) | Models / Datasets | Ti (7M) | S (30M) | M (126M) | L (480M) |
| 1D-CFD (N) | **0.04203** | 0.05715 | 0.06534 | 0.05056 | 1D-DR (N) | 0.02147 | 0.02004 | **0.01951** | 0.02553 |
| 2D-DR (N) | 0.12588 | 0.11843 | 0.13524 | **0.11067** | 1D-BE (N) | 0.20365 | 0.21151 | 0.20735 | **0.20164** |
| 2D-CFD-IC (N) | 0.09657 | 0.13037 | 0.09509 | **0.08567** | 2D-CFD (N) | 0.10084 | 0.10066 | 0.09988 | **0.09019** |
| 2D-SW (N) | 0.00751 | 0.00605 | 0.00603 | **0.00454** | 2D-GSDR (V) | 0.50432 | 0.51025 | 0.49516 | **0.48519** |
| 3D-MHD (V) | 0.32203 | 0.31276 | 0.32156 | **0.28496** | 3D-CFD-Turb (N) | 0.31692 | 0.29181 | 0.29507 | **0.28436** |
| 3D-CFD (N) | 0.09163 | 0.11308 | 0.11541 | **0.07279** | 3D-TGC (V) | 0.53468 | 0.48879 | 0.49386 | **0.47722** |

attention; and POSEIDON (Herde et al., 2024), a hierarchical multiscale operator transformer with shifted-window attention and lead-time-conditioned normalization for continuous-time querying.

**Evaluation Metrics.** We follow the conventions of PDEBENCH (PB) and THE WELL (TW) when reporting errors on spatiotemporal PDE datasets. The *Normalized Root Mean Squared Error* (NRMSE) is the RMSE normalized by the $\ell_2$ norm of the ground truth, making it a *scale-independent* (dimensionless) measure and the default metric in PDEBENCH (Takamoto et al., 2022). In contrast, the *Variance-Scaled RMSE* (VRMSE) normalizes the RMSE by the target's per-snapshot standard deviation (i.e., the square root of its variance), thereby comparing errors to the intrinsic variability of the field. THE WELL adopts VRMSE as a primary metric (Ohana et al., 2024).

**Comparison Caveats.** A central challenge in fairly comparing PDE foundation models is their use of different *pretraining corpora*. In this work, we therefore fine-tune the FMs on the same downstream tasks and interpret the results independently of their pretraining corpus. Another difficulty in comparing MORPH to prior foundation models is the data modality. The existing models are trained on 2D benchmarks and are consequently constrained to 2D inputs, whereas MORPH is *agnostic to data modality*. To ensure fairness, we report head-to-head results only where the data modality aligns with a given baseline, i.e., table entries are marked "−" when a model cannot be evaluated due to such shape constraints. A further complication for fair comparison is the *input context window* employed by prior foundation models. MPP is trained with a 16-step context, while DPOT uses a 10-step context. In line with numerical solvers, we posit that PDE foundation models should initialize from an initial condition (or initial time step) and advance the state autoregressively into the future. Accordingly, we train with a single-step context, $AR(1)$, for all models.

## 4.2 MAIN RESULTS

Our primary experiments are organized in Tables 3,4,5. In these tables, we compare baseline and SOTA models using NRMSE and VRMSE as evaluation metrics.

Table 5: **Full-shot finetuning results for foundation models**: NRMSE (N) and VRMSE (V) of the experiments (lower is better). The *best score* is typeset in **bold**. The *global best* for the dataset is **bold + underlined**. Our foundation models in red. An asterisk (*) on the L model indicates LoRA fine-tuning.

| Models | DPOT-FM | | Poseidon-FM | | MORPH-FM | | | |
|---|---|---|---|---|---|---|---|---|
| Datasets | S (30M) | M (122M) | T (21M) | B (158M) | Ti (7M) | S (30M) | M (126M) | L* (480M) |
| FT sets **(Full-shot)** | | | | | | | | |
| 1D-DR (N) | - | - | - | - | **_0.0008_** | 0.00131 | 0.00173 | 0.00135 |
| 1D-BE (N) | - | - | - | - | **_0.0302_** | 0.0584 | 0.04177 | 0.04123 |
| 2D-CFD (N) | 0.77357 | 0.63255 | 0.55897 | 0.59939 | 0.05401 | 0.05104 | 0.05886 | **0.0423** |
| 2D-GSDR (V) | 0.00749 | 0.00744 | 0.01199 | 0.01416 | 0.00763 | 0.00725 | 0.00518 | **_0.00497_** |
| 3D-CFD-Turb (N) | - | - | - | - | 0.10742 | 0.10205 | **_0.08635_** | 0.09170 |
| 3D-TGC (V) | - | - | - | - | 0.05312 | 0.04136 | 0.03756 | **_0.03246_** |

**Standalone surrogates.**   In Table 3, we compare performance of MORPH standalone surrogates (MORPH-SS) on pretraining and finetuning datasets against existing baseline (FNO and UNets) and SOTA models (MPP and DPOT). The results for baselines and SOTA models are inherited from the respective benchmark sources (Takamoto et al., 2022; Ohana et al., 2024) and articles (McCabe et al., 2024; Hao et al., 2024). Our MORPH-SS-Ti and S models outperform baseline and SOTA models on standalone datasets and remain competitive across the rest. Practically, this makes MORPH a viable standalone surrogate when the target application focuses on a single PDE family.

**Zero-shot performance.**   In Table 4, we report the zero-shot performance of different MORPH foundation models. For this evaluation, we run the pretrained FMs in inference mode on the test sets of their corresponding datasets. We observe that MORPH-FM-L achieves the best performance on 11 out of 12 datasets. These results indicate favorable scaling behavior of MORPH under zero-shot adaptation.

**Full-shot finetuning performance.**   Full-shot fine-tuning results are presented in Table 5. On the 2D datasets, we compare the fine-tuning performance of MORPH-FM to DPOT-FM and Poseidon-FM, and find that our models consistently surpass these existing FMs. Together with Tables 3 and 5, we can quantify the adaptation-efficiency gains of MORPH-FM over MORPH-SS on the fine-tuning benchmarks: MORPH-FM outperforms MORPH-SS on 5 out of 6 fine-tuning datasets.

**Parameter-efficient finetuning with LoRA.**   We investigate fine-tuning the MORPH-FM-L model using low-rank adapters (LoRA). In our setup, only 77M of the 480M parameters comprising LoRA adapters on attention and MLP blocks, positional encodings and layer norms are fine-tuned, which is fewer trainable parameters than in the M model (126M). Notably, MORPH-FM-L matches or outperforms the M model across full-shot fine-tuning (Table 5). To our knowledge, this is one of the first two concurrent studies to apply LoRA to PDE foundation models (see also: Holzschuh et al. (2025)), and our results indicate that LoRA-based fine-tuning is a promising direction given the expanding PDE datasets and increasing model sizes. Future work will examine LoRA-based fine-tuning more extensively, including the effect of adapter rank across different datasets.

**Autoregressive rollouts.**   We have presented autoregressive rollouts results for standalone model and foundation model. Figs. 6 and 7 (Appendix C.3) show rollout results for MORPH-SS-Ti and MORPH-SS-S models on the Shallow Water Equations (SWE) dataset, using the $t=0$ snapshot as input. Across the rollout horizon, MORPH-SS-S exhibits lower error than MORPH-SS-Ti, consistent with the NRMSE reported in Table 3. Both models produce stable multi-step forecasts, exhibiting limited error accumulation and no blow-up for the 10-step horizon. Fig. 3 presents autoregressive rollouts for MORPH-FM-S fine-tuned on the FNS-KF prediction task for $v_y$ ($v_x$ in Appendix C.3 Fig 9). It can be observed that MORPH-FM-S achieves better rollouts than MORPH-FM-Ti (Fig. 8 in Appendix C.3).

### 4.3   SCALING STUDIES

We study scaling behavior in both pretraining and fine-tuning, with respect to both dataset size and model capacity. For pretraining, Fig. 5 (Appendix C.1) reports training and validation losses for MORPH-FM-S (30M parameters) as we vary the number of pretraining trajectories. The results are

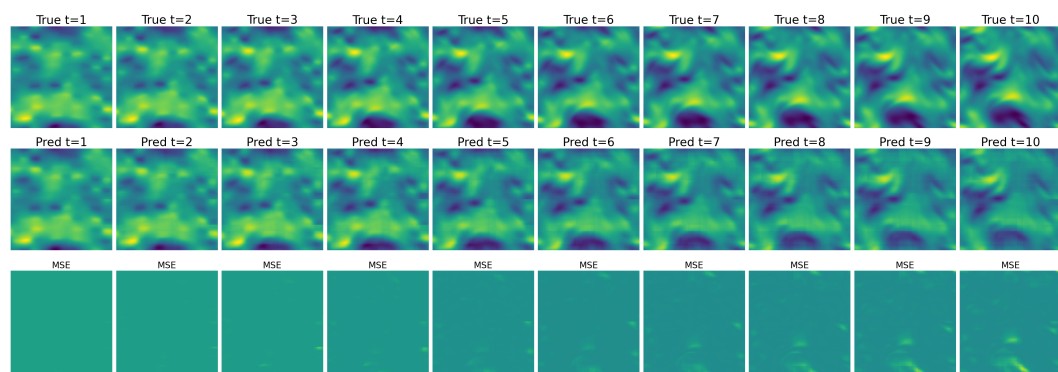

Figure 3: **Finetuning MORPH-FM-S ($\sim$ 30M) for FNS-KF prediction task**: 10-step autoregressive rollouts of $v_y$ with $t = 0$ (initial frame) as input.

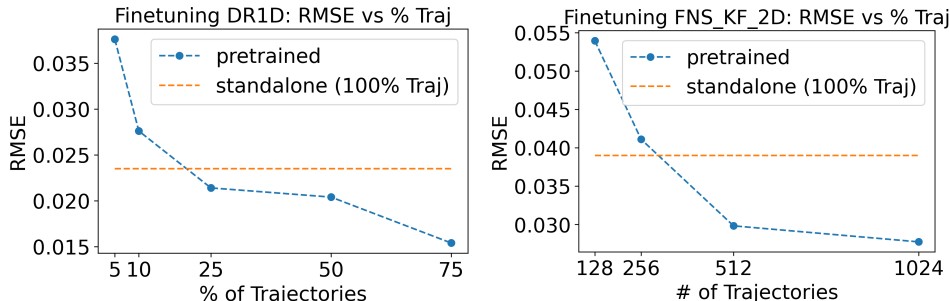

Figure 4: Scaling with respect to finetuning dataset size: (Left:) **MORPH-FM-Ti on 1D-DR**: RMSE vs % different trajectories finetuned for *100 epochs*, (Right:) **MORPH-FM-S on 2D-FNS-KF**: RMSE vs # trajectories of finetuend for *100 epochs*.

shown over the first 10 epochs, during which both losses decrease steadily as the pretraining dataset size increases. For brevity, more details are deferred to Appendix C.1. Turning to model capacity, our largest model (MORPH-FM-L) ultimately saturates at a validation loss of $\sim$0.02 during pretraining, compared to $\sim$0.09 for the Ti/S variants. However, for a fixed training budget of 10 epochs under a single shared protocol (same learning rate, batch size, and weight decay), we do not observe clear gains when scaling from Ti to L. This suggests that mild model-specific hyperparameter tuning and/or extended training is needed for the increased capacity to operate in the regime where performance scales predictably with model size.

**Scaling w.r.t. finetuning dataset size.** Fig. 4 presents a dataset–size scaling study on two downstream tasks (1D-DR and 2D-FNS-KF). We fix the training protocol (optimizer, schedule, epochs, batch size) and vary the *number of fine-tuning trajectories*. We report RMSE as a function of the number of trajectories used to fine-tune MORPH-FM, and compare against the standalone MORPH-SS trained on 100% of trajectories. Errors consistently decrease as dataset size increases. On 1D-DR, the pretrained MORPH-FM-Ti already outperforms the standalone model with only 25% of the trajectories. On 2D-FNS-KF, with $\sim$ 256 trajectories (less than 1% of the full set), MORPH-FM-S surpasses the standalone MORPH-SS-S trained on all trajectories.

**Scaling w.r.t. finetuning model size (across FMs).** To analyze scaling behavior across foundation models of varying capacities, we perform a comparative scaling study on the FNS-KF dataset. We fix the finetuning data size (*128 trajectories, $\leq$1% of full data*) and compute budget (*200 epochs*), and compare the performance of MORPH-FM-S against larger PDE foundation models (DPOT-FM-M and POSEIDON-FM-B). As summarized in Table 8 (Appendix C.1), MORPH-FM-S contains only $\sim$0.25$\times$ the parameters of DPOT-FM-M and $\sim$0.18$\times$ that of POSEIDON-FM-B, yet achieves superior accuracy under identical data- and compute-scarce settings. This experiment high-

lights scaling efficiency, showing that MORPH attains competitive or better performance at significantly smaller model scales. Additional details on the hyperparameter settings and fine-tuning setup are provided in Appendix B.3.2.

### 4.4 ABLATION STUDIES

**Model architecture.** Table 6 summarizes ablations over the convolutional operator, field-fusion module, and attention operator in MORPH. Introducing a lightweight convolutional operator with 8 filters consistently reduces error on 2D/3D benchmarks with only a modest increase in GFLOPs, while having negligible effect on 1D. For field fusion, cross-attention outperforms a simple concat+MLP scheme across datasets, at nearly identical parameter counts and modest additional compute. Among attention variants, axial attention systematically offers the best accuracy–compute trade-off, outperforming both full and sparse self-attention. The gains are most pronounced at larger patch-token counts, where axial factorization yields large FLOP savings while still improving test loss. More details are deferred to Appendix C.2.

Table 6: Ablation studies (Model architecture): Effect of convolution operator, cross-attention field fusion and axial attention on the total number of parameters (in M), floating point operations of the operator (in GFLOPs), and test loss (MSE) for different datasets. ps = patch size, Np - number of patches or tokens.

| Ablations | Variant | MHD3D | | | DR2D | | | CFD1D | | |
|---|---|---|---|---|---|---|---|---|---|---|
| | | Param. | GFLOPs | Loss | Param. | GFLOPs | Loss | Param. | GFLOPs | Loss |
| Conv Filters | 0 (no conv) | 8.29 | 0 | 0.18376 | 7.04 | 0 | 0.00787 | 7.14 | 0 | 0.17185 |
| | **8** | 8.94 | 2.7556 | **0.16851** | 7.96 | 0.11376 | **0.00747** | 8.06 | 0.011 | 0.17348 |
| Field-fusion | Concat & Dense | 8.8 | 0.2013 | 0.17101 | 7.89 | 0.10066 | 0.00827 | 7.99 | 0.05 | 0.18141 |
| | **Cross-attention** | 8.94 | 0.4713 | **0.16851** | 7.96 | 0.16829 | **0.00747** | 8.06 | 0.1178 | **0.17348** |
| | | CFD2D-IC (ps = 8, Np = 4096) | | | CFD2D-IC (ps = 16, Np = 1024) | | | SW2D (ps = 8, Np = 256) | | |
| Attention | Full | 5.78 | 5.9056 | 0.01228 | 14.2 | 75.141 | 0.01208 | 4.67 | 0.6711 | 0.00291 |
| | Sparse | 5.78 | 1.8832 | 0.01324 | 14.2 | 8.6064 | 0.00856 | 4.67 | 0.4372 | 0.00208 |
| | **Axial** | 8.94 | 1.8832 | **0.00812** | 17.3 | 8.6064 | **0.00724** | 7.83 | 0.4372 | **0.00172** |

**Pretraining corpus.** To evaluate the effect of the pretraining set and to disentangle the influence of the dataset from that of the model architecture, we pretrain MORPH-FM-S*(27M) on the same datasets as Poseidon. We then finetune both MORPH and Poseidon on 128 trajectories and test on 240 trajectories for 200 epochs across three downstream tasks: CE-RM, NS-PwC, and NS-SL. The results reported in Table 9 (Appendix C.2) demonstrate that MORPH-FM-S*(27M), with only one-fifth the number of parameters, performs competitively with or even outperforms Poseidon. These findings indicate that the MORPH architecture yields performance gains that are largely independent of the choice of pretraining corpus. More details about this study are provided in Appendix C.2, and full autoregressive rollouts are shown in Figs. 10, 12 and 11 (Appendix C.3).

## 5 CONCLUSION

We introduced MORPH, a modality-agnostic foundation model for PDEs that unifies heterogeneous spatiotemporal data across 1D–3D domains, scalar/vector fields, and mixed resolutions within a single convolutional vision transformer backbone. Our Unified Physics Tensor Format (UPTF-7) standardizes layout across PDE datasets while preserving physical semantics of the observations. By combining component-wise convolutions, inter-field cross-attention, and 4D axial attention, MORPH learns locality, cross-physics dependencies, and long-range spatiotemporal structure in a compute-aware way, and scales across multiple model variants. These design choices, together with parameter-efficient LoRA adapters for finetuning, enable effective transfer from diverse pretraining corpora to downstream tasks. We evaluated MORPH against its standalone version, task-specific baselines and recent PDE foundation models. Beyond prediction accuracy, MORPH highlights pathways toward flexible and scalable PDE foundation models for scientific machine learning.

## REPRODUCIBILITY STATEMENT

All datasets used for pretraining and fine-tuning are publicly available. We will release the full codebase and model checkpoints under an open-source license upon publication.

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

# Appendices

## A  DATA DETAILS

### A.1  BENCHMARK: PDEBENCH

#### A.1.1  1D BURGERS' EQUATION

**Description**: It is a canonical nonlinear advection–diffusion PDE that interpolates between smooth diffusion and shock-forming hyperbolic flow as viscosity decreases. By varying $\nu$, it effectively sweeps a Reynolds-like number and reveals steepening, shock formation, and dissipation—making it a clean test for nonlinearity and shock handling.

**PDE**: $\partial_t u(t, x) + \partial_x \left( \frac{1}{2} u^2(t, x) \right) = \frac{\nu}{\pi} \partial_{xx} u(t, x)$.

**Domain**: $x \in (0, 1),\ t \in (0, 2]$.

**BCs & ICs.** Periodic BCs on $(0, 1)$. Initial condition is a random superposition of sines $u_0(x) = \sum_i A_i \sin(k_i x + \phi_i)$ with uniformly sampled amplitudes/phases/wavenumbers.

**Numerical scheme**: 2nd-order upwind finite difference scheme in space and time.

**Data specifications**: $N = 10{,}000$ trajectories, $T = 201$ time steps, $W = 1024$ spatial points, one scalar field variable $u$. We use dataset corresponding to $\nu = 0.001$.

**Category**: Finetuning

#### A.1.2  1D DIFFUSION-REACTION

**Description**: It combines linear diffusion with a local, potentially rapid (nearly exponential) source term, yielding stiff transients and front-like dynamics. It stresses models' ability to handle disparate time scales and locally amplified errors.

**PDE**: $\partial_t u - \nu\, \partial_{xx} u - \rho\, u(1 - u) = 0$.

**Domain**: $x \in (0, 1),\ t \in (0, 1]$.

**BCs & ICs.** Periodic BCs. ICs drawn from the same random sinusoidal family as Burgers with rectification/normalization for well-posedness.

**Numerical scheme** Finite-difference time–space solver with the piecewise-exact solution (PES) method for the source term part.

**Data specifications**: $N = 10{,}000$ trajectories, $T = 101$ time steps, $W = 1024$ spatial points, one scalar field variable $u$. We use dataset corresponding to $\nu = 0.5, \rho = 1.0$.

**Category**: Finetuning

#### A.1.3  2D DIFFUSION-REACTION

**Description**: The FitzHugh–Nagumo (FHN) system is a prototypical reaction–diffusion equation with applications to modeling biological pattern phenomenon. It consists of two nonlinearly coupled scalar fields i.e., activator and inhibitor.

**PDE**: $\partial_t u = D\nabla^2 u + R(u)$, where , $R_u = u - u^3 - k - v, R_v = u - v$

**Domain**: $(x, y) \in (-1, 1)^2,\ t \in (0, 5]$.

**BCs & ICs.** No–flux (Neumann) BCs. ICs are standard normal random noise.

**Constants**: $k = 5 \times 10^{-3}, D_u = 1 \times 10^{-3}, D_v = 5 \times 10^{-3}$.

**Numerical scheme** Finite-volume in space and classical RK4 in time.

**Data specifications**: $N = 1{,}000$ trajectories, $T = 101$ time steps, $H \times W = 128 \times 128$ spatial points, two scalar field variables $(u, v)$. We use dataset corresponding to $\nu = 0.5, \rho = 1.0$

**Category**: Pretraining

### A.1.4 COMPRESSIBLE NAVIER-STOKES (CNS) SYSTEM

**Description**: The CNS equations describe the motion of Newtonian fluids in regimes where density variations due to pressure are significant. This setting is fundamental in aerodynamics, gas dynamics, aeroacoustics, astrophysical flows, weather and climate dynamics etc. The governing PDEs are conservation laws for mass, momentum, and energy. The CNS solutions may exhibit discontinuities/shocks and turbulent cascades a larger range of multiscale behavior.

**PDE**:

$$\partial_t \rho + \nabla \cdot (\rho \mathbf{v}) = 0,$$
$$\rho(\partial_t \mathbf{v} + \mathbf{v} \cdot \nabla \mathbf{v}) = -\nabla p + \eta \Delta \mathbf{v} + (\zeta + \eta/3)\nabla(\nabla \cdot \mathbf{v}),$$
$$\partial_t \left(\epsilon + \tfrac{1}{2}\rho\|\mathbf{v}\|^2\right) + \nabla \cdot \left[(\epsilon + p + \tfrac{1}{2}\rho\|\mathbf{v}\|^2)\mathbf{v} - \mathbf{v} \cdot \sigma'\right] = 0.$$

where $\rho$ is the density, $\mathbf{u}$ the velocity field, $p$ the pressure, $\epsilon$ the internal energy, $\eta$ the shear viscosity, $\zeta$ the bulk viscosity, and $\sigma'$ the viscous stress tensor.

**BCs & ICs**: Periodic BCs, Random ICs.

**Numerical scheme**: For advection/shock terms: HLLC + MUSCL (robust, shock-capturing, 2nd order). For viscous/diffusion terms: central differencing (smooth, consistent with Navier–Stokes viscosity). More details on the solver Toro et al. (1994); Van Leer (1997).

**Data specifications and category**:

1. 1d CNS (Pretraining): $N = 10{,}000$ trajectories, $T = 100$ time steps, $W = 1024$ spatial points, one vector field ($v$), two scalar fields ($\rho, P$). We use dataset corresponding to $\eta = 10^{-2}, \zeta = 10^{-2}$.

2. 2d CNS (Finetuning): $N = 10{,}000$ trajectories, $T = 21$ time steps, $H \times W = 512 \times 512$ spatial points, one vector field ($v$), two scalar fields ($\rho, P$). We use dataset corresponding to $\eta = 10^{-8}, \zeta = 10^{-8}, M = 0.1$.

3. 3d CNS (Pretraining): $N = 200$ trajectories, $T = 21$ time steps, $D \times H \times W = 128 \times 128 \times 128$ spatial points, one vector field ($v$), two scalar fields ($\rho, P$). We use dataset corresponding to $(\eta, \zeta, M){=}(10^{-8}, 10^{-8}, 0.1)$ and $(10^{-8}, 10^{-8}, 1.0)$.

4. 3d CNS-Turbulent (Finetuning): $N = 500$ trajectories, $T = 21$ time steps, $D \times H \times W = 64 \times 64 \times 64$ spatial points, one vector field ($v$), two scalar fields ($\rho, P$). We use dataset corresponding to $(\eta, \zeta, M) = (10^{-8}, 10^{-8}, 1.0)$.

### A.1.5 2D INHOMOGENEOUS, INCOMPRESSIBLE NAVIER-STOKES

**Description**: The inhomogeneous, forced incompressible Navier–Stokes (INS) system models Newtonian fluids in the low–Mach regime where density is effectively constant. It underpins a wide range of applications, including hydromechanics, environmental and geophysical flows, weather and climate components at resolved scales, and engineered processes such as mixing and cooling.

**PDE**: $\nabla \cdot \mathbf{v} = 0, \rho(\partial_t \mathbf{v} + \mathbf{v} \cdot \nabla \mathbf{v}) = -\nabla p + \eta \Delta \mathbf{v} + \mathbf{u}$, where $\mathbf{v} = (v_x, v_y)$ is velocity, $p$ pressure, $\eta$ the (shear) viscosity, and $\mathbf{u}$ the inhomogeneous forcing.

**Domain**: $\Omega \in [0, 1]^2$.

**BCs & ICs**: Non-periodic Dirichlet (no-slip) BCs. Initial conditions $v_0$ and inhomogeneous forcing parameters $u$ are independently sampled from isotropic Gaussian random fields with truncated power-law power spectra.

**Constants**: Viscosity $\nu = \eta/\rho = 0.01$. Density is treated as constant.

**Numerical scheme**: Finite-difference simulations implemented in PhiFlow (Holl et al., 2020).

**Data specifications**: $N = 80$ trajectories, $T = 1000$ time steps, $H \times W = 512 \times 512$ spatial points, one vector field ($v$), one static vector field ($F$) (not used). We use 4 dataset files.

**Category**: Pretraining

## A.2 BENCHMARK: THE WELL

### A.2.1 2D GRAY-SCOTT DIFFUSION REACTION

**Description**: Two-species reaction–diffusion model whose scalar concentrations vary in space and time. It exhibits rich pattern formations and is used as a canonical test for nonlinear spatiotemporal dynamics and biological morphogenesis (Gray & Scott, 1984).

**PDE**: $\partial_t u = D_u \nabla^2 u - uv^2 + F(1-u); \partial_t v = D_v \nabla^2 v + uv^2 - (F+k)v$.

**Domain**: $\Omega \in [-1,1]^2$.

**BCs & ICs**: Doubly periodic BCs

**Constants**: $\delta_A = 2 \times 10^{-5}, \delta_B = 1 \times 10^{-5}$.

**Numerical scheme**: Implicit-explicit exponential time-differencing RK4 in time and Fourier spectral method in space.

**Data specifications**: $N = 200$ trajectories, $T = 1001$ time steps, $H \times W = 128 \times 128$ spatial points, two scalar field variables $(u, v)$. We use dataset corresponding to "Bubbles" pattern with $f = 0.098, k = 0.057$.

**Category**: Finetuning.

### A.2.2 3D MAGNETOHYDRODYNAMICS

**Description**: A multiphysics dataset coupling fluid dynamics and electromagnetism. MHD is a strongly nonlinear system in which coupled fluid–magnetic dynamics generate broadband turbulence, shocks and compressions, intermittent current sheets, and reconnection, yielding larger dynamics range of spatial and temporal scales. MHD modeling finds applications in understanding solar winds, interstellar medium, magnetospheres, fusion energy devices like tokamaks (Burkhart et al., 2020).

**PDE**: Ideal MHD with isothermal EOS $p = c_s^2 \rho$:

$$\partial_t \rho + \nabla \cdot (\rho \mathbf{v}) = 0, \quad \partial_t (\rho \mathbf{v}) + \nabla \cdot \left[ \rho \mathbf{v}\mathbf{v} + \left( p + \frac{B^2}{8\pi} \right) \mathbf{I} - \frac{1}{4\pi} \mathbf{B}\mathbf{B} \right] = \mathbf{f}, \quad \partial_t \mathbf{B} - \nabla \times (\mathbf{v} \times \mathbf{B}) = 0.$$

**Domain**: Periodic cube.

**BCs & ICs**: Periodic BCs, continuous large-scale solenoidal forcing at $k \approx 2.5$ (box units).

**Constants**: Isothermal sound speed $c_s$. Parameter sweeps over sonic Mach number $M_s \in \{0.5, 0.7, 1.5, 2.0, 7.0\}$ and Alfvénic Mach number $M_A \in \{0.7, 2.0\}$.

**Numerical scheme**: Third-order hybrid ENO shock-capturing scheme (Cho & Lazarian, 2003), isothermal EOS and forced turbulence. The high-res data were generated at $256^3$ and anti-aliased downsampled to $64^3$ for the MHD_64 split.

**Data specifications**: $N = 97$ trajectories, $T = 101$ time steps, $D \times H \times W = 64 \times 64 \times 64$ spatial points, two vector field variables $(v, B)$ and one scalar field $(\rho)$. We use all the files present on the repository.

**Category**: Pretraining.

### A.2.3 3D TURBULENT COOLING GRAVITY

**Description:** Self-gravitating, compressible gas with radiative cooling/heating. It is used to model turbulent molecular clouds and multiphase interstellar medium where cooling drives filament formation and collapse.

**PDE:** Compressible hydrodynamics with gravity and energy source terms (monatomic gas $\gamma = 5/3$):

$$\frac{d\rho}{dt} = -\rho \nabla \cdot \mathbf{v}, \quad \frac{d\mathbf{v}}{dt} = -\frac{\nabla P}{\rho} - \nabla \Phi + \mathbf{a}_{\text{visc}}, \quad \frac{du}{dt} = -\frac{P}{\rho} \nabla \cdot \mathbf{v} + \frac{\Gamma - \Lambda}{\rho},$$

with Poisson gravity through $\Phi$; cooling/heating via metallicity-dependent $\Lambda, \Gamma$.

**Domain:** 3D volume.

**BCs & ICs:** Isolated self-gravitating gas sphere ($10^6 \, M_\odot$) with turbulent velocity spectrum $\propto k^{-4}$;

**Constants:** Adiabatic index $\gamma = 5/3$.

**Numerical scheme:** Density-Independent SPH (DISPH) in `ASURA-FDPS`; radiative processes via tabulated cooling/heating (Hirashima et al., 2023).

**Data specifications:** $N = 20{,}000$ trajectories, $T = 21$ time steps, $D \times H \times W = 64 \times 64 \times 64$ spatial points, one vector field variable ($v$) and three scalar field ($\rho, P, T$). Out of 4 fields, we only take 3 independent fields ($v, P, T$). We use single dataset file corresponding to $T_0 = 10, \rho_0 = 0.445, Z = 0.1$.

**Category:** Finetuning

### A.3 BENCHMARK: PDEGYM

#### A.3.1 FNS-KF

**Description:** Two-dimensional Kolmogorov flow: an incompressible fluid driven by a steady sinusoidal body force.

**PDE:** Forced incompressible Navier–Stokes:

$$\partial_t \mathbf{u} + (\mathbf{u}\cdot\nabla)\mathbf{u} + \nabla p - \nu \, \Delta \mathbf{u} = \mathbf{f}, \qquad \nabla\cdot\mathbf{u} = 0,$$

with time-independent forcing

$$f(x, y) = 0.1 \, \sin\big(2\pi(x + y)\big), \qquad \partial_t f \equiv 0.$$

**Domain:** $D = [0, 1]^2$.

**BCs & ICs:** Periodic boundary conditions. Initial velocity is set exactly as in NS-PwC (Herde et al., 2024) by prescribing a piecewise-constant vorticity $\omega_0$ on a $p \times p$ uniform partition (with $p = 10$) and recovering a divergence-free velocity:

$$\omega_0(x, y) = c_{ij} \quad \text{for } (x, y) \in [x_{i-1}, x_i] \times [y_{j-1}, y_j], \qquad x_i = y_i = \frac{i}{p}, \quad c_{ij} \sim \mathcal{U}[-1, 1].$$

The velocity $\mathbf{u}_0$ is obtained from $\omega_0$ under $\nabla\cdot\mathbf{u}_0 = 0$.

**Constants:** Small effective viscosity from the spectral–hyperviscosity setup, $\nu = 4 \times 10^{-4}$, forcing amplitude 0.1 and wavenumber $2\pi$ along diagonal.

**Numerical scheme:** Pseudospectral Fourier method with artificial (hyper)viscosity and projection to the first N modes using AZEBAN spectral hyperviscosity code.

**Data specifications:** $N = 20{,}000$ trajectories, $T = 21$ time steps, $H \times W = 128 \times 128$, one vector field variables ($v$). We assemble all the '.nc' files in the data directory into a single combined file (Herde et al., 2024). However, we use limited trajectories.

**Category:** Finetuning

### A.4 DATA PROCESSING

**Unified Physics Tensor Format (UPTF-7).** We convert batches into the UPTF-7 format while loading them on-the-fly onto the GPUs. The UPTF-7 layout is defined as $(b, t, F, C, D, H, W) = (\text{batches}, \text{time-steps}, \text{fields}, \text{components}, \text{depth}, \text{height}, \text{width})$, and is used for both the pretraining

and fine-tuned datasets.

$$
\begin{aligned}
\text{1D-CFD} &= (b, t, 3, 1, 1, 1, 1024), & \text{2D-DR} &= (b, t, 2, 1, 1, 128, 128), \\
\text{2D-CFD-IC} &= (b, t, 1, 2, 1, 512, 512), & \text{2D-SW} &= (b, t, 1, 1, 1, 128, 128), \\
\text{3D-MHD} &= (b, t, 3, 3, 64, 64, 64), & \text{3D-CFD} &= (b, t, 3, 3, 128, 128, 128), \\
\text{1D-DR} &= (b, t, 1, 1, 1, 1, 1024), & \text{1D-BE} &= (b, t, 1, 1, 1, 1, 1024), \\
\text{2D-FNS-KF} &= (b, t, 1, 2, 1, 128, 128), & \text{2D-CFD} &= (b, t, 3, 2, 1, 512, 512), \\
\text{2D-GSDR} &= (b, t, 2, 1, 1, 128, 128), & \text{3D-CFD-TURB} &= (b, t, 3, 3, 64, 64, 64), \\
\text{3D-TGC} &= (b, t, 3, 3, 64, 64, 64). &&
\end{aligned}
$$

**Normalization.** We use Reversible Instance Normalization (ReVIN) to counter covariate shift across heterogeneous datasets and training regimes. It was first proposed for time-series forecasting and validated to mitigate distribution shift (Kim et al., 2021). We normalize the data and cache the corresponding means and standard deviations, then use these statistics to exactly denormalize model outputs. Unlike McCabe et al. (2024), which normalizes on-the-fly, we pre-normalize the entire dataset and train on normalized batches, reducing the normalization overhead incurred during training and fine-tuning, and inference.

**Data streaming and sharding setup.** MORPH is designed to handle data heterogeneity and diversity, which introduces system-level challenges. To utilize CPU, GPU, and RAM efficiently, we make several non-traditional choices. One issue is that computing cluster resources impose RAM limits, making it difficult to load all datasets into memory simultaneously. As PDE benchmarks continue to grow, streaming becomes essential for building large PDE foundation models. Accordingly, we stream datasets from storage in chunks, which uses RAM efficiently and scales with the number and size of datasets. Another issue is the wide range of token lengths (128–4096) across pretraining datasets, which leads to varying computational burdens in parallel processing across batches. Instead of a single PyTorch `DataLoader` for all datasets, we use one dataloader per dataset. With six pretraining datasets, we run six DataLoaders. This enables dataset-specific choices of batch sizes and number of workers. Given this multi-dataloader setup, data sharding under `DistributedDataParallel`(DDP) is nontrivial, so we implement custom sharding across workers and ranks.

We deploy six PyTorch `DataLoader` instances and use `DistributedDataParallel` (DDP) to shard data across ranks and workers while streaming. The data chunks are loaded from storage into RAM, locally shuffled, and converted into AR pairs. We first count the total number of autoregressive (AR) samples $T$ across all HDF5 files, then balance the workload across $W$ ranks and $K$ `DataLoader` workers by setting $G = W \cdot K$ and dropping any remainder so $T^*$ is divisible by $G$. Each sub-worker receives exactly $m = T^*/G$ samples and is identified by `my_id = rank · K + worker_id`. As we stream the data in chunks, we maintain a running global index $g$ for every candidate sample; a sub-worker processes a sample iff $g\%G = $ `my_id`. We reseed the RNG each epoch with `base_seed+epoch` and only shuffle within each chunk before forming AR pairs, yielding a single-pass, low-memory pipeline with no duplicate samples and balanced work across all ranks and workers.

---

**Algorithm 1:** Simple Streaming Sharding (DDP ranks × DataLoader workers)

---

**Input:** Ordered files `FILES`, autoregressive window `ar_order`, chunk size, world size $W$,
      `rank`, num workers $K$, `worker_i base_seed`, epoch **Output:** Stream of
      $(X, y)$ pairs for this sub-worker

**Count samples.**
Compute $T = \sum\limits_{f \in \texttt{FILES}} N_{\text{sims}}(f) \cdot \max(0, T_{\text{steps}}(f) - \texttt{ar\_order})$;

**Balance across sub-workers.**
$G \leftarrow W \cdot K; \quad T^* \leftarrow T - (T \bmod G); \quad m \leftarrow T^*/G$;
`my_id` $\leftarrow$ rank $\cdot K +$ `worker_id`; global_idx $\leftarrow 0$;
Seed RNG $\leftarrow$ `base_seed` + epoch;

**Stream and assign.**;
**foreach** $f \in$ `FILES` **do**
    Open $f$ and iterate in chunks of size `chunk_size`;
    For each chunk: shuffle with the epoch seed and prepare AR pairs $(X, y)$;
    **foreach** $(x_i, y_i) \in (X, y)$ **do**
        **if** $global\_idx \geq T^*$ **then**
            **stop**
        **if** $(global\_idx \bmod G) = my\_id$ **then**
            **yield** $(x_i, y_i)$; **if** *this sub-worker has yielded* $m$ *samples* **then**
               **stop**
        global_idx $\leftarrow$ global_idx $+ 1$;

---

# B    Experiment Details

## B.1   Model details

**Model Variants**: We introduce four different variants of MORPH presented in Table 7, based on the number of convolutional filters, dimensions of attention layers, number of attention heads and depth of the transformer.

Table 7: Four Variants of MORPH with the associated hyperparameters

| Size | Conv. filters | Attention dims | Heads | Depth | MLP dims | Model size |
|---|---|---|---|---|---|---|
| MORPH-TI (Tiny) | 8 | 256 | 4 | 4 | 1024 | 7M |
| MORPH-S (Small) | 8 | 512 | 8 | 4 | 2048 | 30M |
| MORPH-M (Medium) | 8 | 768 | 12 | 8 | 3072 | 126M |
| MORPH-L (Large) | 8 | 1024 | 16 | 16 | 4096 | 480M |

**Convolutional operator**: The 3D convolutional operator works on the component dimension of the UPTF-7. By default, we use 8 filters and a 2-layer network with LeakyReLU activations. Inputs with up to `max_in_ch` channels are zero-padded (if needed) and projected to $h=8$ channels by a $1\times1\times1$ convolution (bias-free). We then apply $L$ blocks of $3\times3\times3$ convolution followed by LeakyReLU ($\alpha=0.2$), doubling the channel width each block until reaching $F$ output channels: $h \rightarrow \min(2h, F) \rightarrow \cdots \rightarrow F$. All $3\times3\times3$ layers use padding $= 1$ to preserve spatial size. Unless noted, $F=8$, yielding two convolutional layers in total (the $1\times1\times1$ projection and one $3\times3\times3$ layer).

**Patching**: We partition each sample into non-overlapping patches with patch size 8 along the spatial axes of the tensor $(D, H, W)$. Accordingly, the per-patch shapes are $1\times8\times8$ for 2D data, $1\times1\times8$ for 1D data, and $8\times8\times8$ for 3D data. In our pretraining corpus, the 2D-CFD-IC and 3D-CFD datasets yield the largest number of spatial patches per batch, i.e., `max_patches` = 4096. Practically, the choice of patch size is constrained by available compute. Reducing the patch size increases the sequence length and thus raises memory and runtime costs during training and inference.

**Field-wise cross-attention (field fusion).** After projecting each field patch to a common embedding size (`model_dim`), we form a length-$F$ sequence $\mathbf{X} \in \mathbb{R}^{F \times E}$ per patch and apply a single-layer,

$H$-head cross-attention (default $H$=32) in which the *query* is a learned ($\mathbf{q} \in \mathbb{R}^E$), while $\mathbf{X}$ provides both keys and values. The cross-attention operation performs content-based pooling across the $F$ field variables via a learned query. The time complexity scales linearly with $F$ (a single query attends over $F$ keys). The module naturally accommodates a variable number of fields F at runtime and inference time. We intentionally omit field-wise positional encodings, making the field-fusion module permutation-invariant to the ordering of fields.

Concretely, with head dimension $d_h = E/H$ and per-head projections $W_Q^{(h)}, W_K^{(h)}, W_V^{(h)} \in \mathbb{R}^{E \times d_h}$,

$$\alpha^{(h)} = \text{softmax}\left(\frac{(\mathbf{q}W_Q^{(h)})(\mathbf{X}W_K^{(h)})^{\top}}{\sqrt{d_h}}\right) \in \mathbb{R}^{1 \times F}, \qquad \mathbf{z}^{(h)} = \alpha^{(h)}(\mathbf{X}W_V^{(h)}) \in \mathbb{R}^{1 \times d_h},$$

and the fused representation is

$$\mathbf{z} = \text{Concat}_h(\mathbf{z}^{(h)})\,W_O \in \mathbb{R}^{1 \times E},$$

**Axial attention (factorized 4D space–time).** Given patch embeddings $x \in \mathbb{R}^{B \times t \times N \times E}$ with $N$=$DHW$, we reshape to $x \in \mathbb{R}^{B \times t \times D \times H \times W \times E}$ and replace full space–time self-attention over $L$=$tDHW$ tokens with four 1D multi-head attention (MHA) operations applied along each axis i.e., time ($t$), depth ($D$), height ($H$), and width ($W$). For each axis we fold the remaining axes into the batch, attend along the axis length, invert the reshape, and use a residual sum:

$$\tilde{x}_{b,d,h,w}^{(t)} = \text{MHA}_t\big(x_{b,\cdot,d,h,w}\big) \in \mathbb{R}^{t \times E},$$
$$\tilde{x}_{b,t,h,w}^{(D)} = \text{MHA}_D\big(x_{b,t,\cdot,h,w}\big) \in \mathbb{R}^{D \times E},$$
$$\tilde{x}_{b,t,d,w}^{(H)} = \text{MHA}_H\big(x_{b,t,d,\cdot,w}\big) \in \mathbb{R}^{H \times E},$$
$$\tilde{x}_{b,t,d,h}^{(W)} = \text{MHA}_W\big(x_{b,t,d,h,\cdot}\big) \in \mathbb{R}^{W \times E},$$
$$y = x + \tilde{x}^{(t)} + \tilde{x}^{(D)} + \tilde{x}^{(H)} + \tilde{x}^{(W)},$$

followed by flattening back to $\mathbb{R}^{B \times t \times N \times E}$. We adopt a pre-norm Transformer block: LN $\rightarrow$ axial attention $\rightarrow$ residual, then LN $\rightarrow$ MLP (GELU, Dropout) $\rightarrow$ residual. The temporal branch is enabled only when $t$>1, which supports an autoregressive curriculum (stage 1 uses spatial-only AR(1); stage 2 enables AR(2:$k$)).

*LoRA parameterization.* All attention projections (Q, K, V, O) and the two MLP linear layers are equipped with low-rank adapters; when rank $r$=0 they reduce to plain linear maps. For input $x$ and base weight $W_0$, a LoRA-enhanced linear layer produces

$$y = xW_0^{\top} + \frac{\alpha}{r}\big(xA^{\top}\big)B^{\top} + b,$$

with $A \in \mathbb{R}^{r \times \text{in}}$, $B \in \mathbb{R}^{\text{out} \times r}$, scaling $\alpha/r$, and optional dropout $p$ on the LoRA path. This adds $r(\text{in} + \text{out})$ parameters per linear while leaving the base weights reusable/freezable.

*Complexity.* Full space–time attention costs $\mathcal{O}\big((tDHW)^2\big)$ in sequence length. Axial factorization reduces this to

$$\mathcal{O}\big(tDHW\,(t + D + H + W)\big),$$

since each axis attends over its own length while other dimensions are folded into the batch, yielding substantial savings in compute and memory without sacrificing global receptive field.

*Defaults.* We use same attention heads (Table 7) across axes, dropout $p$ inside attention and MLP, and LoRA rank $r$ with scaling $\alpha$ (LoRA inactive when $r$=0).

**Positional encodings**: Let a learned absolute table $\mathbf{P} \in \mathbb{R}^{\text{max\_ar} \times \text{max\_patches} \times E}$ provide time–patch embeddings, and let token tensors be $x \in \mathbb{R}^{B \times t \times n \times E}$. At run time we construct $\widehat{\mathbf{P}} \in \mathbb{R}^{t \times n \times E}$ to match the current autoregressive context $t$ and patch count $n$, and add it to $x$ (broadcast over the batch). We use two different types of positional encoding.

*ST-Bilinear*: We resample $\mathbf{P}$ *jointly* over time and patches using bilinear interpolation:

$$\widehat{\mathbf{P}}_{\tau,\pi} = \sum_{i=1}^{\text{max\_ar}} \sum_{j=1}^{\text{max\_patches}} w_{\tau,i}^{(t)}\,w_{\pi,j}^{(n)}\,\mathbf{P}_{i,j}, \qquad \widehat{\mathbf{P}} \in \mathbb{R}^{t \times n \times E},$$

with interpolation weights $w^{(t)}$ and $w^{(n)}$ derived from the continuous reindexing of $[1, \texttt{max\_ar}] \to [1, t]$ and $[1, \texttt{max\_patches}] \to [1, n]$ (no temporal slicing). This preserves smooth variation across both axes and supports arbitrary $(t, n)$ without changing parameter count. We employ this variant for the **L** model with $\texttt{max\_ar} = 16$.

*S-Linear & T-Slice*: We *slice* the first $t$ time steps and *linearly* resample only along the patch axis:

$$\widehat{\mathbf{P}}_{\tau,\pi} = \sum_{j=1}^{\texttt{max\_patches}} w_{\pi,j}^{(n)} \mathbf{P}_{\tau,j}, \qquad \tau \in \{1, \ldots, t\}, \ t \leq \texttt{max\_ar},$$

yielding $\widehat{\mathbf{P}} \in \mathbb{R}^{t \times n \times E}$. This avoids temporal smoothing, preserves the semantics of discrete AR time steps, and is lighter-weight computationally. We use this variant for the **Ti/S/M** models with $\texttt{max\_ar} = 1$.

Both encodings are *absolute* and enable variable $(t, n)$ by resampling, with an $\mathcal{O}(t\,n\,E)$ application cost. ST-Bilinear provides smooth extrapolation in time and space—beneficial for long AR horizons (L model). S-Linear/T-Slice enforces a strict AR horizon ($t \leq \texttt{max\_ar}$) and avoids temporal aliasing—well-suited to short-context regimes (Ti/S/M). We apply dropout after constructing $\widehat{\mathbf{P}}$ and then add it to token embeddings prior to attention.

## B.2 PRETRAINING SETTINGS

**Balanced task sampling.** We train MORPH on multiple pretraining datasets with varying numbers of trajectories per dataset. We perform balanced task sampling to avoid imbalanced learning and catastrophic forgetting. Each dataset is assigned a sampling weight inversely proportional to its number of trajectories, so datasets with fewer trajectories are randomly sampled more frequently. In our corpus, the 3D-MHD dataset has the fewest trajectories and is therefore sampled more often than the remaining five. For multi-dataset training, each dataset $i$ is assigned a sampling weight inversely proportional to its number of trajectories $N_i$, i.e., $w_i \propto 1/N_i$, and the weights are normalized so that $\sum_i w_i = 1$. The resulting empirical per-batch sampling probabilities are approximately: 3D-MHD: 0.31, 2D-CFD-IC: 0.19, 3D-CFD: 0.18, 2D-DR: 0.12, 2D-SW: 0.12, and 1D-CFD: 0.08.

**LoRA layers.** Low-Rank Adaptation (LoRA) is used for fine-tuning LLMs where the pretrained model weights are frozen while introducing trainable rank decomposition matrices into the transformer layers, dramatically reducing task-specific parameters during finetuning (Hu et al., 2022). The motivation is supported by evidence that over-parameterized networks possess low intrinsic dimensionality, suggesting that task-specific updates lie in a low-rank subspace (Aghajanyan et al., 2020). Mathematically, for a frozen weight $W_0 \in \mathbb{R}^{d \times k}$, LoRA parameterizes the update as

$$\Delta W = BA, \qquad B \in \mathbb{R}^{d \times r}, \ A \in \mathbb{R}^{r \times k}, \ r \ll \min(d, k),$$

yielding the adapted weight

$$W' = W_0 + \frac{\alpha}{r} BA,$$

where $\alpha$ is a scaling factor. Only $A$ and $B$ are trained while $W_0$ remains fixed. We apply LoRA to the dense projections in attention and MLP blocks of the large model, with default ranks $r_{\text{attn}} = 16$ and $r_{\text{mlp}} = 12$, respectively.

**Compute infrastructure.** We ran experiments across multiple GPU configurations, depending on availability and the computational requirements of each job. The model was executed on personal workstations as well as on cluster configurations consisting of multiple nodes and GPUs. This also demonstrates that the model scales from a single GPU to distributed settings with minimal configuration changes.

1. **Standalone surrogates.** All 12 MORPH standalone models were trained on either $2\times$ A100 GPUs (40 GB each) or $2\times$ A6000 GPUs (48 GB each).

2. **Small foundation models.** MORPH-FM-TI (7M) and MORPH-FM-S (30M) were trained on a single node with $4\times$ A100 GPUs (40 GB each).

3. **Medium/Large foundation models.** MORPH-FM-M (126M) and MORPH-FM-L (480M) were trained on two nodes, each equipped with $8\times$ H100 GPUs (80 GB each).

4. **Fine-tuning.** Fine-tuning was performed on either $1\times$ H100 (MORPH-FM-M and MORPH-FM-L) or $2\times$ A100/A6000 (MORPH-FM-TI and MORPH-FM-S).

5. **Comparisons.** MPP, DPOT, and POSEIDON were fine-tuned on $1\times$ H100.

**Other details.**

- **Data splits**: Train/Val/Test: 0.8/0.1/0.1
- **Data pipeline**: Custom chunked streaming with deterministic sharding across workers.
- **Training duration**: Standalone and fine-tuned models are trained for approximately 100–150 true epochs. For foundation models, we pretrain for 200K gradient steps for the TI and S configurations, and 100K steps for the M and 125K steps for the L variants.
- **Batch size**: Dataset heterogeneity necessitates dataset-specific batch sizes, implemented via per-dataset `DataLoader`. Accounting for our continuous data-streaming pipeline and targeting high GPU utilization on a $2\times$ A100/A6000 setup, we use the following per-GPU batch sizes:

$$1D\text{-}CFD = 128, \quad 2D\text{-}DR = 64, \quad 2D\text{-}CFD\text{-}IC = 16, \quad 2D\text{-}SW = 64$$
$$3D\text{-}MHD = 16, \ 3D\text{-}CFD = 4, \ 1D\text{-}DR = 384, \ 1D\text{-}BE = 384, \ 2D\text{-}FNS\text{-}KF = 64$$
$$2D\text{-}CFD = 8, \quad 2D\text{-}GSDR = 64, \quad 3D\text{-}CFD\text{-}TURB = 16, \quad 3D\text{-}TGC = 16.$$

When training with more than two GPUs or L model, we scale these per-GPU batch sizes by $0.25\times$.

- **Optimizer**: We use AdamW (Loshchilov & Hutter, 2017) with a learning rate of $10^{-3}$ and weight decay $10^{-2}$. For standalone CFD datasets, we use $10^{-4}$. The learning-rate schedule is `ReduceLROnPlateau` with a decay factor of $0.5$ and a patience of $5$ epochs, applied after a 20-epoch warmup.
- **Early stopping**: We trigger early stopping if the validation loss fails to improve for more than 10 epochs. In practice, early stopping never activates when pretraining the foundation models.

### B.3    FINETUNING SETTINGS

#### B.3.1    MORPH MODELS

**Hyperparameters.**    We use the same normalization and effective batch size as above. Fine-tuning runs for 100–150 true epochs with early stopping, and the learning rate is scheduled with `ReduceLROnPlateau`. We optimize with AdamW. We use a single PyTorch `DataLoader` per dataset. We support both single- and multi-GPU training via `DataParallel`.

**Fine-tuning levels.**    MORPH supports four levels of fine-tuning (level-1 through level-4). Level-1 applies LoRA-based fine-tuning for the MORPH-FM-L model. We set the LoRA ranks on the linear layers in the axial-attention and MLP blocks via arguments. In level-2, we additionally unfreeze the encoder (convolutional, projection, and cross-attention layers). In level-3, we also unfreeze the decoder's projection layer. Level-4 fine-tunes all parameters. For the TI, S, and M models, we use level-4. For level-1 LoRA layers, we use a learning rate of $1e{-}3$ and weight decay $0.0$. For level-4, we use a default learning rate of $1e{-}4$ and weight decay $0.0$.

#### B.3.2    EXISTING SOTA MODELS

To contextualize MORPH against recent PDE foundation models, we fine-tune two Poseidon variants (Poseidon-T (21M) and Poseidon-B (158M) (Herde et al., 2024)) and two DPOT variants (DPOT-S (30M) and DPOT-M (122M) (Hao et al., 2024)). Results are presented in Tables 5, 8 and 9.

We use the official repositories: Poseidon `camlab-ethz/poseidon` at commit `b8fa28f` (retrieved 2025-08-01), and DPOT `HaoZhongkai/DPOT` at commit `dcd2f9a` (retrieved 2025-08-04), with default settings unless noted.

For single-step prediction, we initialize from the released pretrained weights and fine-tune for 100 epochs on the full training split before evaluating on the entire test split (Table 8). The *full-shot* setting in Table 5 evaluates single-step prediction $t \to t+1$ given only the state at $t$.

For the rollout experiment (Tables 8 and 9), we fine-tune on the first 128 trajectories for 200 epochs and evaluate on the last 240 trajectories of the FNS-KF subset of PDEBench, reporting average MSE and RMSE over 20-step rollouts (Takamoto et al., 2022). Throughout, all inputs are at $128{\times}128$ spatial resolution per channel, and we apply the data normalization procedures specified by each repository.

**DPOT.** DPOT is pretrained with a 10-step temporal context at $128{\times}128$ resolution. To adapt DPOT for single-step $t \to t+1$ prediction (Table 5), we follow the input-padding strategy analogous to Herde et al. (2024), feeding 10 copies of the state at time $t$ to satisfy the context window:

$$\underbrace{[x_t, \ldots, x_t]}_{10 \text{ copies}} \longmapsto \widehat{x}_{t+1}.$$

For rollouts (Table 8), when predicting early steps $k+1 \leq 10$, we left-pad the available prefix $[x_0, \ldots, x_k]$ with $x_0$ to length 10; e.g., to predict $x_4$,

$$[\text{ts}_0, \text{ts}_0, \text{ts}_0, \text{ts}_0, \text{ts}_0, \text{ts}_0, \text{ts}_0, \text{ts}_1, \text{ts}_2, \text{ts}_3] \longmapsto \widehat{\text{ts}}_4.$$

For $k+1 > 10$, DPOT consumes the most recent 10 ground-truth states during fine-tuning and the most recent 10 predicted states during rollout evaluation. We use DPOT's repository defaults (AdamW with a One-Cycle learning-rate schedule; no input normalization), and fine-tune with base learning rate $1 \times 10^{-3}$ and batch size 32.

**POSEIDON.** Poseidon takes a single input frame and a selectable $\Delta t$ and predicts $t + \Delta t$. For the single-step experiments (Tables 5 and 9), we fine-tune Poseidon with $\Delta t{=}1$ on all $t \to t+1$ pairs in the training set. For full-trajectory prediction (Table 8), we adopt the repository's default $\Delta t$ fine-tuning schedule with max_num_time_steps = 7 and time_step_size = 2, which enables multi-step forecasting. Results shown for Poseidon in Table 8 are that of Poseidon's direct (non-autoregressive) prediction, where the model directly predicts every time-step in the trajectory without feeding any predictions back into the model. 2D-CFD data in Table 5 is of $(512 \times 512)$ resolution. To align with Poseidon's pretrained weights, we downsample these images to $128 \times 128$ with area averaging, run inference/training at $128 \times 128$, then upsample back to $(512 \times 512)$ with bilinear interpolation for evaluation against ground truth. All Poseidon models were fine-tuned with AdamW (LR $1 \times 10^{-4}$), batch size 40, cosine LR schedule, and gradient clipping (max grad norm = 5.0).

# C ADDITIONAL AND EXTENDED RESULTS

## C.1 SCALING STUDIES

**Scaling w.r.t. pretraining dataset size.** The scaling experiments are conducted on four H100 GPUs using Distributed Data Parallel (DDP). We use the per-GPU batch sizes (global batch size divided by 4) described in Sec. B.2. To vary the effective dataset size, we subsample the training data by randomly retaining a fraction $p\%$ of the mini-batches produced by the training dataloader and processing only these mini-batches in the model's forward pass. Fig. 5 presents the study.

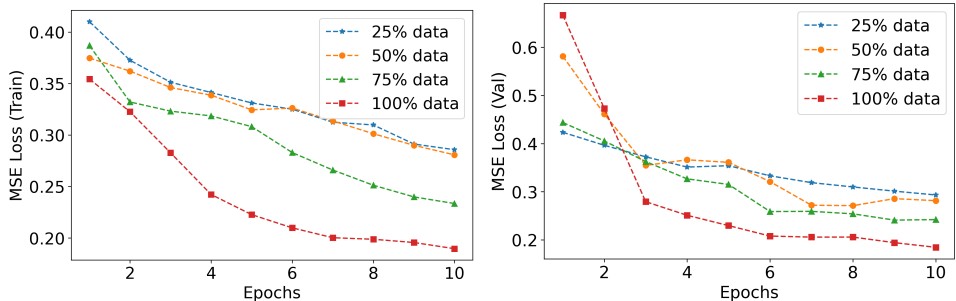

Figure 5: Scaling studies: Data-level scaling for MORPH-FM-S model

**Scaling w.r.t. finetuning model size (across FMs).** We perform full fine-tuning of MORPH-FM-S using a learning rate of 1e-4, batch size 40, and weight decay 1e-4. We employ a ReduceLROn-Plateau scheduler with a wait of 5 epochs and early stopping with a patience of 10 epochs. The hyperparameter settings and fine-tuning setup for Poseidon and DPOT are provided in Appendix B.3.2. Table 8 shows the comparison.

Table 8: Comparisons of PDE foundation models for FNS-KF prediction task with *128 ($\leq 1\%$) finetuning trajectories* and *200 epochs*: MSE and RMSE on test set (lower is better) for DPOT-FM-M vs Poseidon-FM-B vs MORPH-FM-S.

| Models
Metrics | DPOT-FM-M
122M | Poseidon-FM-B
158M | **MORPH-FM-S**
**30M** |
|---|---|---|---|
| MSE | 0.0301 | 0.0017 | **0.00162** |
| RMSE | 0.176 | 0.0412 | **0.0401** |

## C.2 ABLATION STUDIES

**Model architecture.** Table 6 evaluates three key design choices in MORPH i.e., the component-wise convolutional preprocessor, the field-fusion module, and the attention operator. We report parameter count (M), operator FLOPs (GFLOPs), and test MSE on representative datasets (MHD3D, DR2D, CFD1D, CFD2D-IC, and SW2D; here $ps$ denotes patch size and $N_p$ the number of patches/tokens). The ablation studies are performed on 4x H100s GPUs with same hyperparameters.

*Convolutional preprocessor.* Comparing "0 (no conv)" to the 8-filter variant shows that adding a lightweight convolutional operator consistently reduces error on 3D and 2D problems with modest compute overhead: on MHD3D the loss drops from $0.18376$ to $0.16851$ (about $8.3\%$) at the cost of $2.76$ GFLOPs, and on DR2D from $0.00787$ to $0.00747$ (about $5.1\%$) with only $0.11$ additional GFLOPs. For CFD1D the effect is small and slightly negative ($0.17185 \rightarrow 0.17348$), indicating that the convolutional inductive bias is most beneficial for higher-dimensional settings.

*Field-fusion.* For field fusion, cross-attention is compared against a simple concat+dense baseline. Cross-attention consistently improves accuracy with negligible parameter overhead and a moderate increase in FLOPs: on MHD3D the loss decreases from $0.17101$ to $0.16851$ ($\approx 1.5\%$), on DR2D from $0.00827$ to $0.00747$ ($\approx 9.7\%$), and on CFD1D from $0.18141$ to $0.17348$ ($\approx 4.4\%$), while

GFLOPs roughly double but remain small in absolute terms. This supports the use of content-aware attention for fusing multiple fields instead of static mixing.

*Attention operator.* The bottom block compares axial, full, and sparse self-attention for different token counts. Axial attention offers the best accuracy–compute trade-off and its advantage grows with sequence length. On CFD2D-IC with $ps=16$ and $N_p=1024$, axial attention achieves a loss of 0.00724 using 8.61 GFLOPs, whereas full attention reaches 0.01208 at 75.14 GFLOPs (about $8.7\times$ more compute and $\approx 40\%$ higher error). For $ps=8$, $N_p=4096$, axial is again superior (0.00812 vs. 0.01228) while being $\approx 3.1\times$ cheaper in FLOPs, and a similar pattern holds on SW2D where axial is $\approx 1.5\times$ cheaper and $\approx 41\%$ lower in loss than full attention. Compared to sparse attention, axial has essentially the same FLOPs but reduces loss by 15-40% across datasets. These trends are consistent with the theoretical complexity reduction from $O((tDHW)^2)$ for full attention to $O(tDHW\,(t+D+H+W))$ for axial factorization, and motivate axial attention as a near–Pareto-optimal choice for large spatiotemporal token grids.

**Pretraining corpus.** We pretrained MORPH-FM-S (30M) from scratch on the same pretraining corpus as Poseidon i.e., NS-Sines, NS-Gaussians, CE-RP, CE-CRP, CE-KH, CE-Gauss Herde et al. (2024) for 100 epochs. We employ a ReduceLROnPlateau scheduler with a wait of 5 epochs and early stopping with a patience of 10 epochs. We then finetune MORPH on 128 trajectories and test on 240 trajectories for 200 epochs across three downstream tasks: CE-RM, NS-PwC, and NS-SL. We use a learning rate of 1e-4, batch size 40, and weight decay 1e-4, early stopping with patience of 25 during the finetuning. The hyperparameter settings for Poseidon are presented in Appendix B.3.2. Table 8 shows the comparison. Full autoregressive rollouts are shown in Figs. 10, 12 and 11.

Table 9: Ablation studies (Pretraining corpus): MORPH-FM-S*(27M) and POSEIDON-FM-B are finetuned with *128 trajectories* for *200 epochs*: Rollout MSE on test set (lower is better) average over 240 trajectories.

| FT datasets Models | CE-RM | NS-PwC | NS-SL |
|---|---|---|---|
| Poseidon-FM-B (158M) | 0.4181 | **0.0004** | 0.0163 |
| **MORPH-FM-S*** (27M) | **0.09044** | 0.00408 | **0.0159** |

## C.3 AUTOREGRESSIVE ROLLOUTS

### C.3.1 STANDALONE SURROGATE

Figs. 6 and 7 show 10-step autoregressive rollouts for MORPH-SS-TI and MORPH-SS-S on the Shallow Water Equations (SWE) dataset, using the $t=0$ snapshot as input. Across the rollout horizon, MORPH-SS-S exhibits lower error than MORPH-SS-TI, consistent with the NRMSE reported in Table 3. Both models produce stable multi-step forecasts, exhibiting limited error accumulation and no blow-up for the 10-step horizon.

### C.3.2 FINETUNED FOUNDATION MODELS

Figs. 8 and 9 present 10-step autoregressive (AR) rollouts for MORPH-FM-TI and MORPH-FM-S fine-tuned on the forced incompressible Navier–Stokes with Kolmogorov forcing (FNS-KF) dataset, initialized from the t=0 snapshot. Over the 10-step horizon, MORPH-FM-S achieves consistently lower error than MORPH-FM-TI. Both models remain stable under rollouts with limited error accumulation and no concerning instability across the rollout.

### C.3.3 FINETUNED FOUNDATION MODEL (PRETRAINED WITH PDEGYM)

**NS-SL.** Fig. 10 presents full autoregressive (AR) rollouts for MORPH-FM-S*(27M), pretrained on PDEGym (NS-Sines, NS-Gaussians, CE-RP, CE-CRP, CE-KH, CE-Gauss) and subsequently fine-tuned on *128 trajectories of NS-SL* and evaluated on 240 trajectories over 200 epochs. We demonstrate stable rollouts with limited error accumulation even with a compact 27M-parameter MORPH model. These results suggest that MORPH can serve as an efficient foundation model, accurately capturing the underlying dynamics while remaining computationally lightweight.

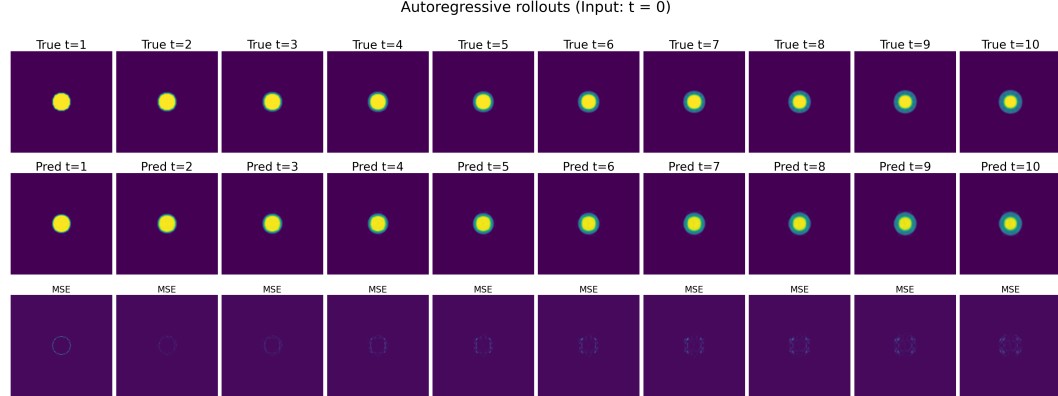

Figure 6: **MORPH-SS-Ti inference**: 10-step autoregressive rollouts for Shallow Water Equations (SWE) with $t = 0$ (initial frame) as input.

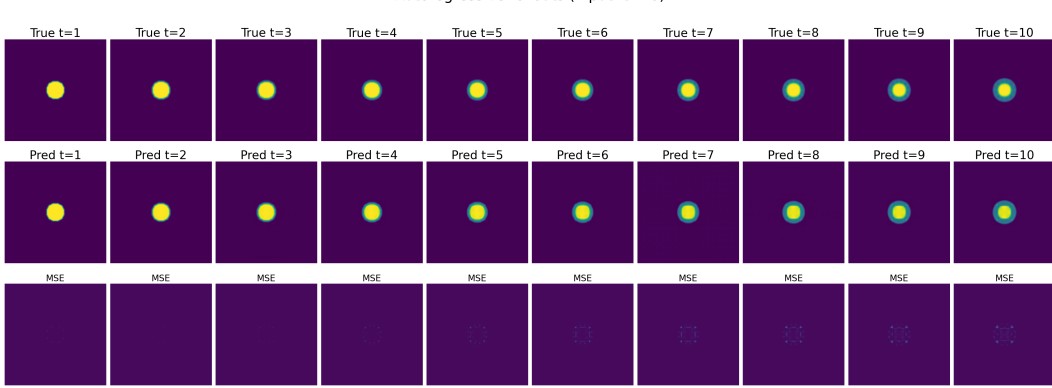

Figure 7: **MORPH-SS-S inference**: 10-step autoregressive rollouts for Shallow Water Equations (SWE) with $t = 0$ (initial frame) as input

**NS-PwC.** Fig. 11 presents full autoregressive (AR) rollouts for MORPH-FM-S*(27M), pretrained on PDEGym (NS-Sines, NS-Gaussians, CE-RP, CE-CRP, CE-KH, CE-Gauss) and subsequently fine-tuned on *128 trajectories of NS-PwC* and evaluated on 240 trajectories over 200 epochs. We obtain stable rollouts with limited error accumulation even using a compact 27M-parameter MORPH model. This indicates that MORPH can function as an efficient foundation model, capturing the essential dynamics while remaining computationally efficient.

**CE-RM.** Fig. 12 presents full autoregressive (AR) rollouts for MORPH-FM-S*(27M), pretrained on PDEGym (NS-Sines, NS-Gaussians, CE-RP, CE-CRP, CE-KH, CE-Gauss) and subsequently fine-tuned on *128 trajectories of CE-RM* and evaluated on 240 trajectories over 200 epochs. We observe stable rollouts, but note missing higher spectral components and finer details toward the end of the rollout. This behavior is expected given the relatively small model size (27M parameters) and the limited number of fine-tuning trajectories (128).

CURRENT LIMITATIONS

MORPH is pretrained primarily on fluid-related datasets. However, a general-purpose PDE foundation model should ultimately encompass a broader range of physics, including elasticity problems such as fracture mechanics, wave propagation in solids, and fluid–structure interaction problems. At present, MORPH is restricted to a maximum of 500M parameters. With access to a larger and more diverse pretraining corpus, we expect that scaling the model to a few billion parameters will be necessary to fully exploit the available data. Moreover, the convolutional preprocessing and patch-

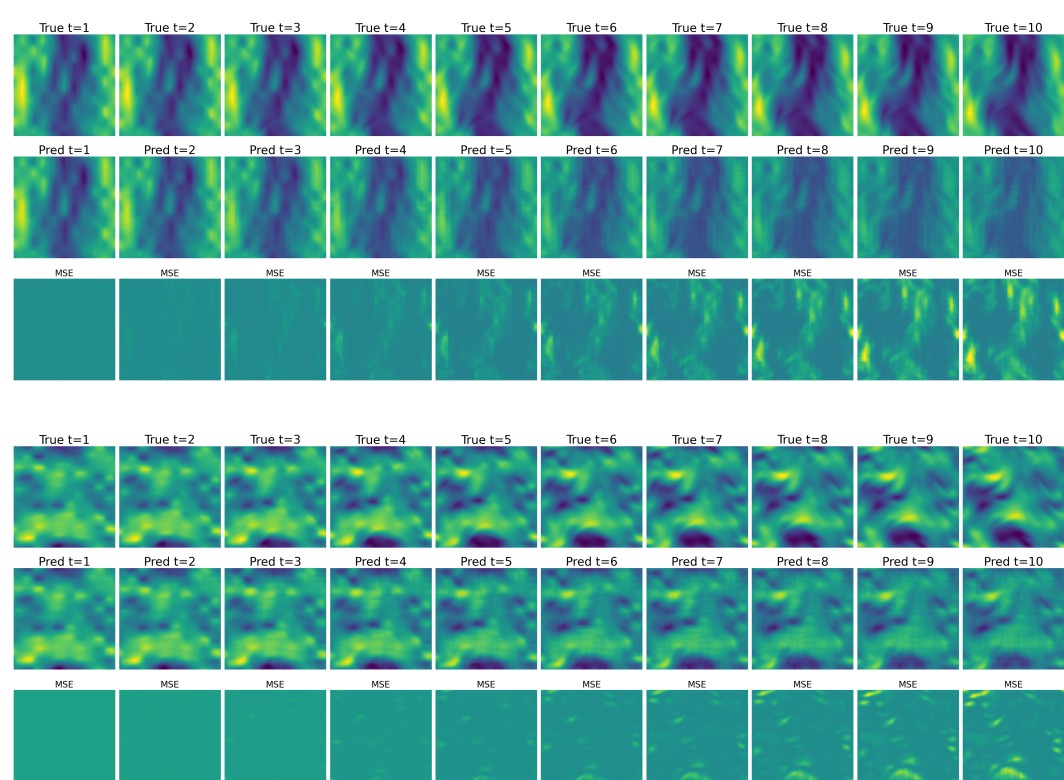

**Figure 8: Finetuning MORPH-FM-Tɪ (∼ 7M) for FNS-KF prediction task**: 10-step autoregressive rollouts for $v_x$ and $v_y$ with $t = 0$ (initial frame) as input.

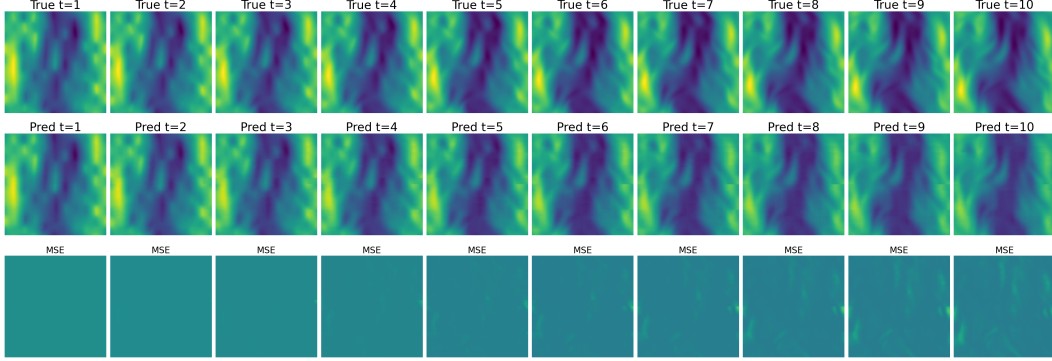

**Figure 9: Finetuning MORPH-FM-S (∼ 30M) for FNS-KF prediction task**: 10-step autoregressive rollouts for $v_x$ with $t = 0$ (initial frame) as input.

ing scheme is designed for regular grids, which limits direct applicability to unstructured meshes or irregular geometries. Finally, UPTF-7 employs scalar-to-vector lifting, which introduces additional batch-level memory and computational overhead.

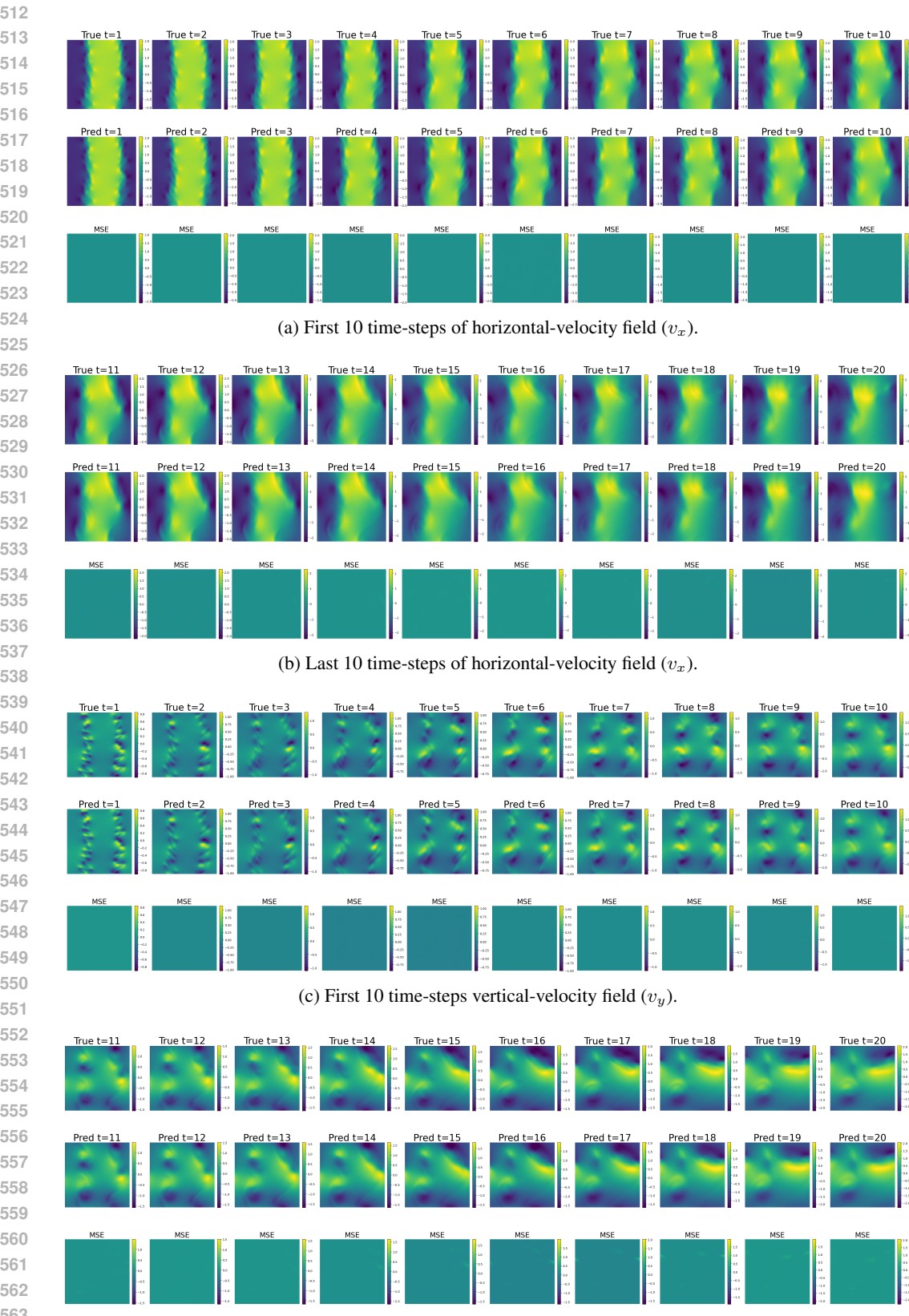

(a) First 10 time-steps of horizontal-velocity field ($v_x$).

(b) Last 10 time-steps of horizontal-velocity field ($v_x$).

(c) First 10 time-steps vertical-velocity field ($v_y$).

(d) Last 10 time-steps vertical-velocity field ($v_y$).

Figure 10: Full autoregressive rollouts for MORPH-FM-S*(27M) finetuned for **NS-SL** prediction task with **128 trajectories**, tested on 240 trajectories for 200 epochs

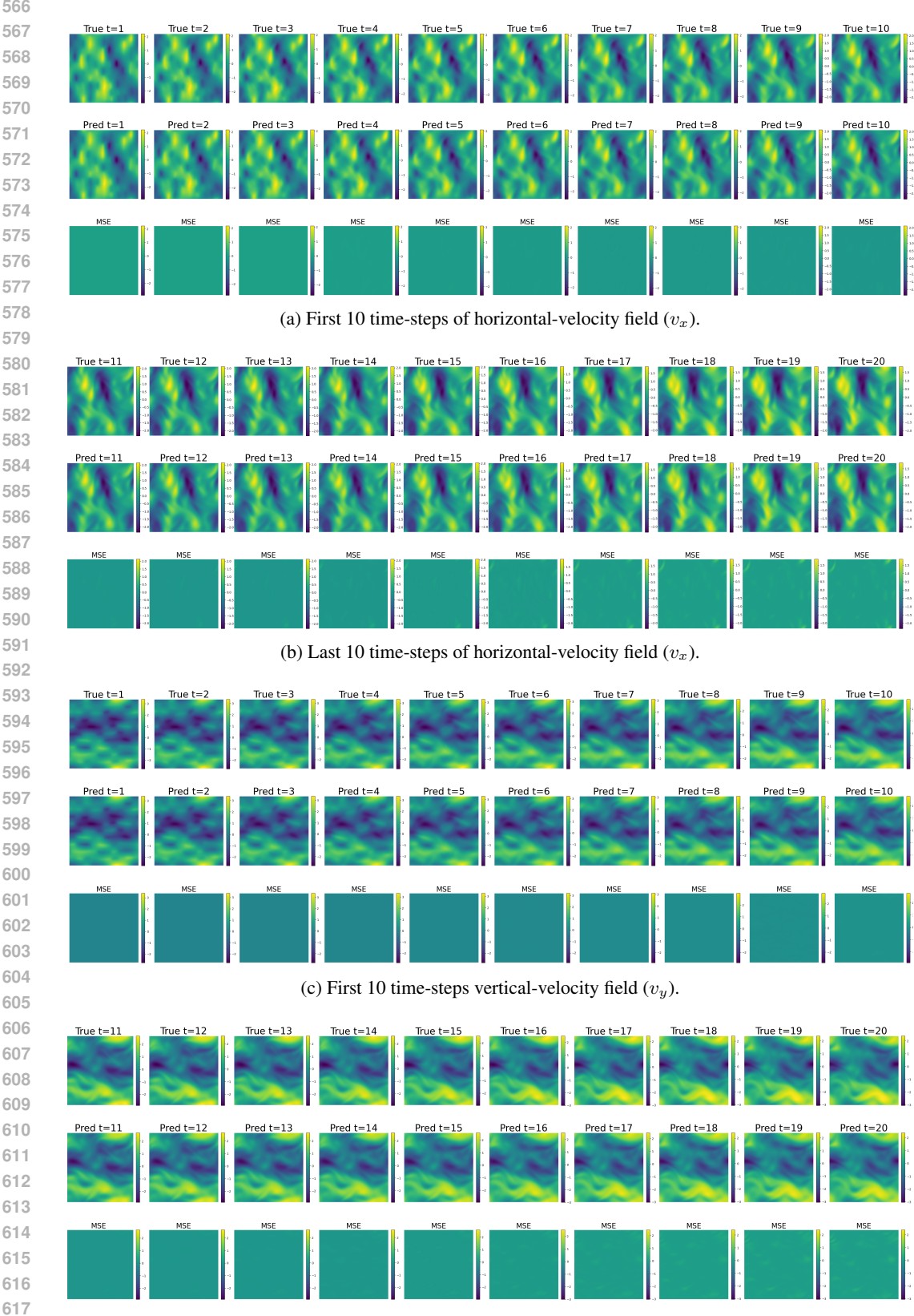

(a) First 10 time-steps of horizontal-velocity field ($v_x$).

(b) Last 10 time-steps of horizontal-velocity field ($v_x$).

(c) First 10 time-steps vertical-velocity field ($v_y$).

(d) Last 10 time-steps vertical-velocity field ($v_y$).

Figure 11: Full autoregressive rollouts for MORPH-FM-S*(27M) finetuned for **NS-PwC** prediction task with **128 trajectories**, tested on 240 trajectories for 200 epochs

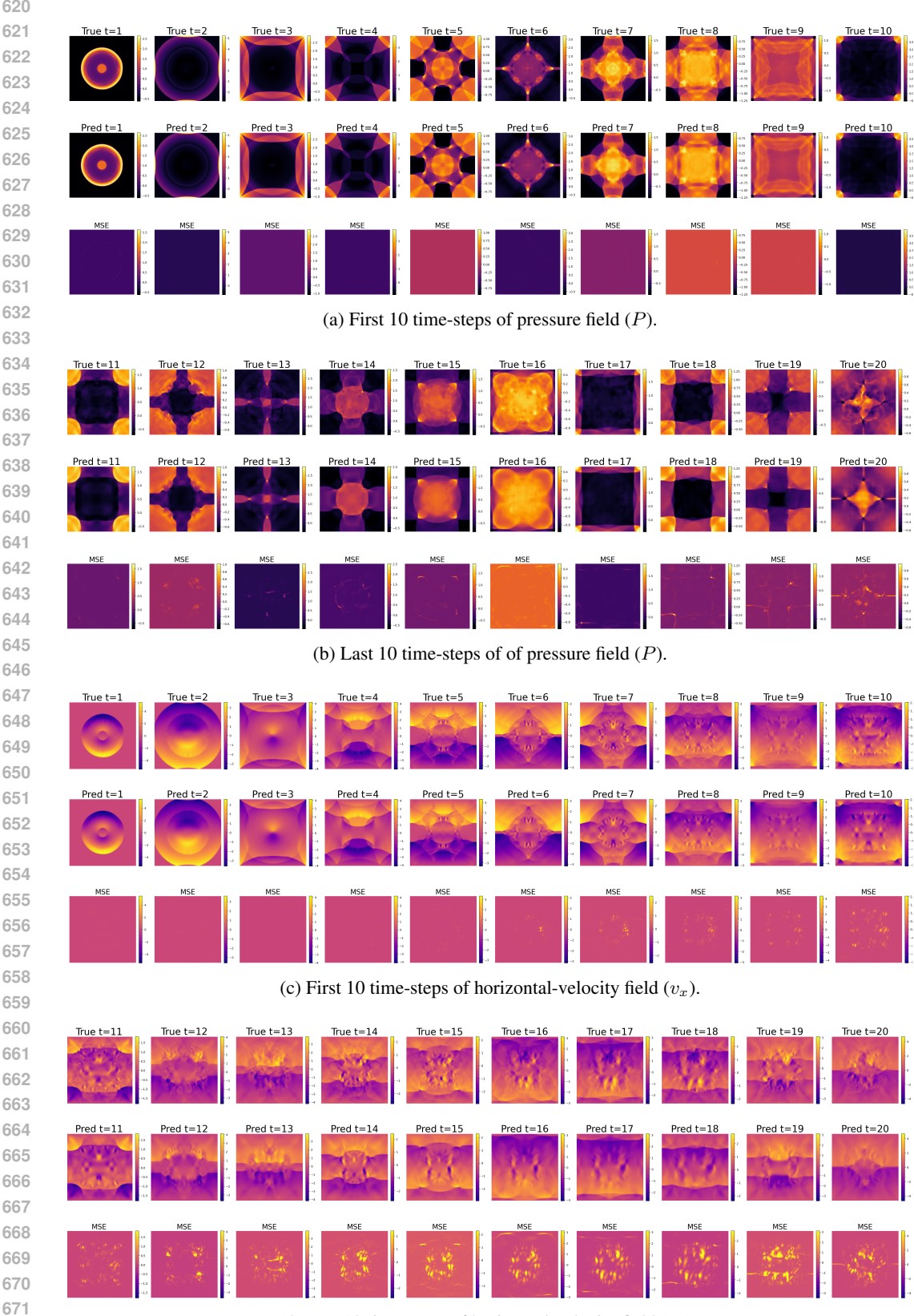

(a) First 10 time-steps of pressure field ($P$).

(b) Last 10 time-steps of of pressure field ($P$).

(c) First 10 time-steps of horizontal-velocity field ($v_x$).

(d) Last 10 time-steps of horizontal-velocity field ($v_x$) .

Figure 12: Full autoregressive rollouts for MORPH-FM-S*(27M) finetuned for **CE-RM** prediction task with **128 trajectories**, tested on 240 trajectories for 200 epochs