# OpenReview forum: "MORPH: Shape-agnostic PDE Foundation Models"
_ICLR.cc/2026/Conference — ICLR 2026 Conference Withdrawn Submission_

### Official Review · Reviewer_53go · 2025-10-25

**Soundness:** 2
**Presentation:** 2
**Contribution:** 2
**Rating:** 4
**Confidence:** 4

**Summary:**

The paper presents a new autoregressive foundation Model called MORPH, which is able to be pretrained on datasets with different numbers of spatial dimensions and fields. The model is based on a vision transformer and employs patching and axial dimension. The experiments show that the model, with the used pretraining datasets, can beat models trained from scratch and other foundation models.

**Strengths:**

1. Paper is generally well-written and easy to understand.
2. Showed the advantage of using LoRA for PDE foundation models.
3. Model architecture that can deal with different amounts of spatial dimensions as well as fields.

**Weaknesses:**

1. The main paper is too technical. It should focus more on the presented architecture and further experiments. The method section starts with a discussion of the data format. The introduced data format, seems to be however, not different from the standard format implicitly used in all PDE papers (besides maybe interpreting scalar fields and vector fields in the same format). Hence, it would be good to only mention the chosen input format shortly. Similarly, the discussion of the dataloader and whether it is better to apply normalization on the data directly or on the fly could be shortened or omitted in the main part.  Instead, it might also be worthwhile to add ablation and scaling experiments from the appendix to the main part.
2. While the paper is evaluated against other foundation models, the other foundation models are not pretrained with the same datasets. It would still be necessary to evaluate against other PDE foundation models when all have the same pretraining data. Otherwise, we cannot evaluate if MORPH generalizes better than the other models. Of course, Morph is highlighted as a model that can learn across a number of spatial dimensions. Hence, it would make sense to have an experiment where MORPH and  Poseidon, for example, are pretrained on the same 2D data, and another one where you add the 3D and 1D data to show how much MORPH can profit from the other data sources.
3. The second experiment mentioned in 2 would also show if MORPH can really profit from including other-dimensional data. This is also missing in the paper right now in general. It is only shown that MORPH is better when being pretrained vs. from scratch in general, but the pretrained set contains datasets with all spatial dimensions. Having an experiment showing that pre-training with 2D data helps on 3D fine-tuning sets would be very insightful.
3. "Shape" as a word can be confusing since it could also be referring to airfoil shapes in the PDE world, for example, and not necessarily to the tensor shape.

**Questions:**

1. How is the patching done? Do you have a patch encoder for each number of spatial dimensions to project down to the same size across spatial dimensions?
2. What do you mean by partial observability in line 43? How does MORPH approach partial observability?
3. What do you mean by attention "chaos" in line 99?

---

> ### Author Response · Authors · 2025-11-21
> **Reply to reviewers' comment: Weakness section (point 1)**
>
> Thank you for investing enough time to provide guidelines for enhancing the paper quality.
>
> 1a. Reviewer comment: The main paper is too technical. It should focus more on the presented architecture and further experiments. The method section starts with a discussion of the data format. The introduced data format, seems to be however, not different from the standard format implicitly used in all PDE papers (besides maybe interpreting scalar fields and vector fields in the same format). Hence, it would be good to only mention the chosen input format shortly. Similarly, the discussion of the dataloader and whether it is better to apply normalization on the data directly or on the fly could be shortened or omitted in the main part. Instead, it might also be worthwhile to add ablation and scaling experiments from the appendix to the main part.
>
> Authors' response:
>
> 1. We agree with the reviewer on the paper reconfiguration. We will shift section 3.2 and 4.2, which talks about normalization, dataloaders, pretraining setup: task sampling, LoRA layers, compute infra, settings to the appendix section.
>
> 2. Along with this, we bring the results from appendix section C to the main paper.
>
> 3. As suggested by you, we will add ablation studies to the main paper.
>
> 4. We have performed data-size scaling for two finetuning datasets in Figs. 2 and 3 (Appendix). These figures will be moved to the main paper.
>
> **Planned Updates**: With your suggestions, we will update the paper with
> 1. Reconfigure the paper for better readability.
> 2. Add ablation studies to show the effect of different architecture choice on the accuracy, parameters and flops.
> 3. Add more scaling studies to the paper.
>
> Please let us know if we are not understanding your pinpointed concerns. We are taking to address the concerns with our best efforts.

---

> ### Author Response · Authors · 2025-11-21
> **Reply to reviewers' comment: Weakness section (point 2, 3)**
>
> 2. Reviewer comment: While the paper is evaluated against other foundation models, the other foundation models are not pretrained with the same datasets. It would still be necessary to evaluate against other PDE foundation models when all have the same pretraining data. Otherwise, we cannot evaluate if MORPH generalizes better than the other models. Of course, Morph is highlighted as a model that can learn across a number of spatial dimensions. Hence, it would make sense to have an experiment where MORPH and Poseidon, for example, are pretrained on the same 2D data, and another one where you add the 3D and 1D data to show how much MORPH can profit from the other data sources.
>
> 3. Reviewer comment: The second experiment mentioned in 2 would also show if MORPH can really profit from including other-dimensional data. This is also missing in the paper right now in general. It is only shown that MORPH is better when being pretrained vs. from scratch in general, but the pretrained set contains datasets with all spatial dimensions. Having an experiment showing that pre-training with 2D data helps on 3D fine-tuning sets would be very insightful.
>
> Authors comments:
> 1. Central contribution clarification:
> We would like to reiterate that our key contribution lies in MORPH’s ability to jointly handle 1D, 2D, and 3D datasets within a unified framework: a novel direction for PDE foundation models. Because of this design goal, our pretraining corpus necessarily included datasets across different dimensionalities and physical modalities (scalar/vector fields). In contrast, models such as MPP, DPOT, and Poseidon are primarily trained and evaluated on 2D datasets. Therefore, a one-to-one comparison across 1D–3D is not feasible. To maintain fairness, we conduct direct head-to-head comparisons on 2D datasets (a common ground) and provide additional evaluations on 1D and 3D datasets to demonstrate MORPH’s broader generalization scope.
>
> 2. On pretraining corpus mismatch:
> We agree with the reviewer that differing pretraining datasets can complicate direct comparisons. Our evaluation strategy follows established practice in prior PDE foundation model literature, where models pretrained on different corpora are compared on different fine-tuning tasks (e.g., DPOT vs. MPP; Poseidon vs. DPOT). This approach assesses generalization and transfer, even though it is not an entirely “apples-to-apples” comparison. We will acknowledge this limitation explicitly in our Section 4.3. Thanks for pointing out this issue.
>
> 3. Perspective on “foundation” model comparisons:
> We wish to highlight a broader conceptual point, the notion of a “foundation model” can become ambiguous if each model is restricted to its own narrow pretraining domain. We believe the essence of a PDE foundation model lies in capturing shared representations across diverse physical domains, not in matching specific datasets. Hence, we adopted a comparison framework independent of pretraining sets, consistent with prior works such as MPP, DPOT, and Poseidon.
>
> 4. Feasibility of same-data retraining experiments:
> We did not conduct same pretraining data comparisons (e.g., retraining Poseidon, DPOT, or MPP on identical 2D datasets) due to their prohibitive computational cost. Training multiple large-scale foundation models from scratch across multiple PDE datasets would demand extensive resources and divert the focus of the paper. Moreover, MORPH was explicitly designed to handle heterogeneous 1D/2D/3D datasets, as represented in benchmarks like PDEBench, PDEGym, and The Well. We view this model has the ability to integrate across multiple modalities, which is central to scaling laws (the more diverse the physics, the stronger the generalization).
>
> 5. Demonstrating from 2D to 3D transfer:
> We agree with the reviewer point of view of specifically highlighting "Having an experiment showing that pre-training with 2D data helps on 3D fine-tuning sets would be very insightful."
>
> --> Planned updates:
> 1. Clarify baseline variability (Sec. 4.3):
> We will add a short subsection noting that existing foundation models are typically pretrained on differing corpora, and we will discuss how this affects fairness and interpretation in cross-model comparisons (what can and cannot be concluded from non-identical pretraining sets).
>
> 2. Targeted 2D -> 3D transfer study:
> If full-scale 2D to 3D evaluation is not feasible before the deadline, we will include a reduced-compute proxy (e.g., lower resolution or smaller model capacity) with the same controls and report it transparently as preliminary evidence.

---

> > ### Comment · Reviewer_53go · 2025-11-24
> >
> > Thank you for your response.
> >
> > Regarding the pretraining corpus mismatch: While I see that training foundation models is costly, I think this is still crucial for a meaningful comparison. I acknowledge that MORPH has the advantage over Poseidon to be able to deal with 1D and 3D data as well.
> >
> > However, it cannot be ruled out that your fine-tuning tasks are closer to your pretraining tasks than they are to DPOT or Poseidon pretraining tasks. Both in the sense of physics as well as in settings as the number of time steps or the resolution.  If, for each new PDE foundation model, we present a new set of pretraining tasks and a new set of fine-tuning tasks, we can't judge the effect of the different pretraining data vs the effect of the new model. Of course, these are expensive experiments, but they are still necessary for a fair and scientific comparison.

---

> > > ### Author Response · Authors · 2025-12-03
> > > **Reply to reviewer comment: Regarding the pretraining corpus mismatch**
> > >
> > > **Reviewer comment:**
> > >
> > > Regarding the pretraining corpus mismatch: While I see that training foundation models is costly, I think this is still crucial for a meaningful comparison. I acknowledge that MORPH has the advantage over Poseidon to be able to deal with 1D and 3D data as well.
> > >
> > > However, it cannot be ruled out that your fine-tuning tasks are closer to your pretraining tasks than they are to DPOT or Poseidon pretraining tasks. Both in the sense of physics as well as in settings as the number of time steps or the resolution. If, for each new PDE foundation model, we present a new set of pretraining tasks and a new set of fine-tuning tasks, we can't judge the effect of the different pretraining data vs the effect of the new model. Of course, these are expensive experiments, but they are still necessary for a fair and scientific comparison.
> > >
> > > **Authors' response:**
> > > In response to the reviewers’ emphasis on this issue, we added a new experiment in Ablation Studies (Pretraining Corpus) (Sec. 4.4) specifically designed to address this comment.
> > >
> > > *Write-up*: To evaluate the effect of the pretraining set and to disentangle the influence of the dataset from that of the model architecture, we pretrain MORPH-FM-S*(27M) on the same datasets as Poseidon. We then finetune both MORPH and Poseidon on 128 trajectories and test on 240 trajectories for 200 epochs across three downstream tasks: CE-RM, NS-PwC, and NS-SL. The results reported in Table 9 (Appendix C.2) demonstrate that MORPH-FM-S*(27M), with only onefifth the number of parameters, performs competitively with or even outperforms Poseidon. These findings indicate that the MORPH architecture yields performance gains that are largely independent of the choice of pretraining corpus. More details about this study are provided in Appendix C.2, and full autoregressive rollouts are shown in Figs. 10, 12 and 11 (Appendix C.3).

---

> ### Author Response · Authors · 2025-11-24
> **Reply to reviewer comment: Regarding the pretraining corpus mismatch**
>
> Reviewer comment: Regarding the pretraining corpus mismatch: While I see that training foundation models is costly, I think this is still crucial for a meaningful comparison. I acknowledge that MORPH has the advantage over Poseidon to be able to deal with 1D and 3D data as well.
>
> However, it cannot be ruled out that your fine-tuning tasks are closer to your pretraining tasks than they are to DPOT or Poseidon pretraining tasks. Both in the sense of physics as well as in settings as the number of time steps or the resolution. If, for each new PDE foundation model, we present a new set of pretraining tasks and a new set of fine-tuning tasks, we can't judge the effect of the different pretraining data vs the effect of the new model. Of course, these are expensive experiments, but they are still necessary for a fair and scientific comparison.
>
> Authors response:
>
> - We thank the reviewer for the thoughtful feedback. We agree that pretraining and finetuning corpus alignment is central to a fair evaluation of PDE foundation models; we treat it as an open limitation. Given compute constraints, we cannot retrain DPOT/Poseidon on identical pretraining sets, but **we will add the following partial evidence: [FNS-KF case shifted from appendix to the main paper + new scaling + new ablations + new transfer study].**
>
> - **Finetuning MORPH-S (30M) with FNS-KF task proves out-of-distribution generalization against larger DPOT and Poseidon**: This task comes from PDEGym and it's a finetuning task for Poseidon as well. MORPH has not seen any pretraining tasks coming from PDEGym. We compare DPOT's (122M) and Poseidon's (158M) model against our MORPH-30M model. All of the models: DPOT, Poseidon and MORPH uses only 128 finetuning trajectories and finetuned only for 200 epochs. This particular experiment is highlighted in section 4.4 (main results), passage "data and compute-scarce finetuning", Table 4 appendix. We demonstrate that our small MORPH model outperforms larger prior PDE foundation models under this data- and compute-scarce regime.
>
> **Authors' corrections in the paper**:
> 1. Clarify comparisons: We will explicitly document, in the main text and appendix, the pretraining and fine-tuning task sets and data shapes for MORPH, DPOT, and Poseidon, and we will discuss how these choices may affect transfer to downstream tasks.
> 2. Add new evidences in the paper: New scaling studies, New ablation studies, New transfer study.
>
> We have incorporated a new set of results and restructured the paper to improve readability and clarity. Please see the revised manuscript and our detailed rebuttals, which we hope address your concerns.

---

> ### Author Response · Authors · 2025-11-26
> **Reply to reviewers questions**
>
> Reviewer comments:
> Questions:
> 1. How is the patching done? Do you have a patch encoder for each number of spatial dimensions to project down to the same size across spatial dimensions?
> 2. What do you mean by partial observability in line 43? How does MORPH approach partial observability?
> 3. What do you mean by attention "chaos" in line 99?
>
> Authors'response:
> 1. We use a single patch-embedding mechanism shared across 1D/2D/3D. There is no separate patch encoder per spatial dimensionality.
> - Starting from the UPTF-7 layout $(B,T,F,C,D,H,W)$, we first apply a 3D field-wise convolution over components $C$, obtaining $(B*T*F,filters,D,H,W)$.
> - For the AR(1) setting, we then perform field-wise spatial patching over $(D,H,W)$, which yields patches of shape $(p_D,p_H,p_W)$ and indices $(N_d,N_h,N_w)$. This gives a tensor of shape $(B,F,N_d,N_h,N_w,p_D,p_H,p_W,filters)$, which we reshape to $(B,F,N_p,features)$ with $(N_p = N_d N_h N_w)$.
> - A shared linear projector maps this to $(B,F,N_p,dims)$.
> - Finally, inter-field cross-attention fuses the $F$ fields into a single representation per patch, resulting in $(B,N_p,dims)$ (i.e., $F$ is reduced to 1).
>
> 2. By “partial observability,” we mean that available data provide only limited, heterogeneous views of the physical state: datasets may be 1D/2D/3D, often from simulations with modeling assumptions and numerical errors, while real experiments are frequently 1D.
>
> 3. We originally used “attention chaos” as a loose analogy to “polynomial chaos expansion” to describe attention computed over multiple axes (fields and spatial dimensions: F, D, H, W). Because this term is uncommon and insufficiently justified in the paper, we will replace it with a clearer phrase e.g., “multi-axis attention over (F, D, H, W)” and add a one-sentence definition for readability. Thank you for pointing this out. We will improve the readability and remove the loose analogy.

---

### Official Review · Reviewer_hKzL · 2025-10-29

**Soundness:** 3
**Presentation:** 2
**Contribution:** 2
**Rating:** 2
**Confidence:** 4

**Summary:**

The paper introduces MORPH, a PDE foundation model which is targeted towards heterogeneous data, covering 1D, 2D and 3D at different resolutions. MORPH uses a unified physics tensor format for processing 1D-3D data with different fields. MORPH employs spatial + temporal axial attention as well as a inter-field cross-attention operation. Experiments include comparisons against standard benchmarks and recent foundation-like models such as DPOT, MPP and Poseidon in zero-shot and full-shot settings.

**Strengths:**

- Network architectures for PDEs that can be trained on a combination of 1D, 2D and 3D data with different fields and discretizations is a very timely and relevant problem
- In principle, the proposed network architecture combined with adequate positional embeddings makes a lot of sense and is a sound solution to this problem

**Weaknesses:**

- I am skeptical about the main experiment. For the "standalone surrogate" part, FNO and UNet are standard baselines. What specific U-Net architecture was used? U-Nets are often misrepresented in PDE benchmarks, because more modern variants are not used (e.g. any recent U-Net architecture in a high-impact paper on generative modeling for images). Other sota architectures (MPP, DPOT) are not trained on all datasets for the standalone surrogate (did you use the metrics from previous papers or retrained the models). The metrics seem very noisy; sometimes the smaller models (Ti, S) perform better, sometimes the larger networks (M, L). I would expect the scaling law to hold (larger models perform better), which was also shown in other PDE foundation-like model papers. Did you use the EMA weights for evaluation?

- Continuing on the main experiment, but the "foundation models" part; DPOT and Poseidon were each pretrained on different pretraining datasets (the ones from the original paper) than MORPH? In this case, that would make the comparison difficult, since the pretraining datasets would be another crucial factor for the performance. Also, I am not sure if I would agree that when MORPH is trained on all pretraining datasets, then evaluating them on each of them individually can be considered a "zero-shot" setting. If all foundation models were pretrained on the same datasets, that would make the comparison much more useful, because then it would be possible to assess more clearly how well the transfer from pretraining datasets to finetuning datasets works.

- The main paper contains a lot of technical material that can be moved to the appendix in my opinion to make space for more relevant ablation studies and experiments. For example, details about the data stream and sharding setup as well as  the compute infrasctructure.

- While 3D datasets are considered, the resolutions are not very high. I expect the MORPH to scale well to high resolutions due to the axial attention, however I would appreciate a detailed model comparison regarding GFLOPs, throughput and VRAM requirements of the different networks.

- The architecture of MORPH seems to be very similar to MPP (axial attention over space+time). Can you clarify the differences and if they are very similar, do an ablation study for every modification? I would also appreciate an ablation study on different design choices of the network.

- The results on autoregressive rollouts are not comprehensive; are the models stable for more than 10 autoregressive steps? The dynamics in Figures 4 to 7 do not show much change over time. Why didn't you show rollouts for 3D? Those are most interesting as they are most challenging.

- It's not true that "this is the first study to apply LoRA to PDE foundation models", see [1]

- The word "shape-agnostic" made me first believe the paper proposes a model that works on arbitrary geometries; the current title is overselling it to me as the model is still limited to regular grids.

[1] https://openreview.net/pdf?id=3BaJMRaPSx

**Questions:**

- How is the input history of 10 previous states in DPOT-FM handled in the zero-shot setting?
- Have you considered any methods like Patch n' Pack https://arxiv.org/pdf/2307.06304 or others for training on different datasets and resolutions?
- Have you looked at the performance of MORPH when changing the resolution of a specific pretraining dataset?
- I did not understand if you are using ST-Bilinear or S-Linear & T-Slice for MORPH? Have you considered relative positional encodings, e.g. RoPE? Do you think this makes sense?
- The information density of a patch with patch size 8 is much lower for 1D than for 3D data. Is this a problem?
- Do you use any additional conditioning information like a class label for each dataset? Would this be helpful?

---

> ### Author Response · Authors · 2025-11-21
> **Reply to reviewer comments: Weakness section (point 1)**
>
> Thank you for your careful review and the time you invested. We respond to each of your points below and will revise the manuscript to improve clarity, methodology, and presentation.
>
> 1) Reviewer comment: I am skeptical about the main experiment. For the "standalone surrogate" part, FNO and UNet are standard baselines. What specific U-Net architecture was used? U-Nets are often misrepresented in PDE benchmarks, because more modern variants are not used (e.g. any recent U-Net architecture in a high-impact paper on generative modeling for images).
>
> Authors' response:
>
> - **FNO/UNet benchmarks are used widely and in prior works as well**: We agree that substantial advances beyond FNO and U-Net have appeared. We nevertheless include FNO and U-Net because they remain widely used, open-source, canonical baselines in common benchmarks (e.g., PDEGym, PDEBENCH, The Well). Using these baselines allows a transparent, reproducible comparison against models that many readers know.
>
> - Prior works (e.g., MPP, DPOT, Poseidon) also report results against these same benchmark baselines, so our choice aligns with established practice. Our intent is to position our method relative to both long-standing baselines and recent SOTA, not to exclude newer approaches.
>
> 2) Reviewer comment: “Other SOTA architectures (MPP, DPOT) are not trained on all datasets for the standalone surrogate (did you use the metrics from previous papers or retrain the models?).”
>
> Authors' response:
>
> - For standalone surrogate results, we report the **published numbers for MPP and DPOT from their original papers**.
>
> - **Our central contribution is the ability to handle 1D, 2D, and 3D datasets within a unified framework**, which is not investigated in any PDE foundation models. Therefore, we were bound to use different modalities (1D--3D datasets, each with different types of scalar/vector fields) in pretraining and finetuning sets than MPP and DPOT. Because MPP and DPOT primarily report on 2D, a one-to-one comparison across 1D and 3D is not available. Accordingly, we perform direct head-to-head comparisons on shared 2D datasets (a level playing field), and we highlight our additional results on 1D/3D to demonstrate the broader scope of our approach.
>
> 3) Reviewer comment:
> The metrics seem very noisy; sometimes the smaller models (Ti, S) perform better, sometimes the larger networks (M, L). I would expect the scaling law to hold (larger models perform better), which was also shown in other PDE foundation-like model papers. Did you use the EMA weights for evaluation?
>
> Authors' response:
>
> - Thank you for raising this. The variance is higher than we would like. Two main factors contributed:
>
> 1. Heterogeneous training budgets and resources. The Ti/S models were trained for roughly 2× more gradient steps than the M/L models and, due to availability, on different compute (Sec. 4.2). The unequal budgets inflate variance and complicate scale-to-scale comparisons (details in App. B.3).
>
> 2. Hyperparameter coupling with scale. Learning rate, batch size, and related settings were not fully normalized across Ti/S/M/L.
>
> - Since submission, we have extended training for M/L models and observed improved numbers. More importantly, in the revision we will re-run the scaling study on a level playing field.
>
> - We have performed data-size scaling for two finetuning datasets in Figs. 2 and 3 (Appendix). These figures will be moved to the main paper. We will perform more scaling studies.
>
> 3. We also did not use EMA for evaluation.
>
> --> **Steps taken to improve the paper (based on the comments)**:
> 1. Presenting a section for scaling studies in the updated paper.
> 2. Reconfigured the paper to show results in the main paper and push pretraining setup and data preprocessing details in the appendix.
>
> We appreciate your feedback and are happy to incorporate any additional comparisons or clarifications that would make the evaluation more balanced and useful to readers.

---

> ### Author Response · Authors · 2025-11-21
> **Reply to reviewer comments: Weakness section (point 2)**
>
> 4) Reviewer comment:
> Continuing on the main experiment, but the "foundation models" part; DPOT and Poseidon were each pretrained on different pretraining datasets (the ones from the original paper) than MORPH? In this case, that would make the comparison difficult, since the pretraining datasets would be another crucial factor for the performance.
>
> Author Response:
>
> **Prior works also compare on finetuning sets**: Thank you for highlighting this important point we agree that differences in pretraining corpora complicate direct comparisons. Our evaluation followed common practice in the literature (e.g., DPOT compares with MPP; Poseidon compares with DPOT), where foundation models pretrained on different sets are compared on the same fine-tuning benchmarks.
>
> More broadly, we share the concern that, without a common pretraining corpus, the term “foundation model” can become ambiguous. In this submission we follow established precedent comparing models on shared fine-tuning tasks irrespective of pretraining data and we will present this explicitly as a limitation and a call for community standardization.
>
> Concretely, we will:
>
> - Highlight this aspect in Section 4.3, along with models and evaluation metrics where we discuss other nuances of fair comparisons.
>
> 5) Reviewer comment:
> Also, I am not sure if I would agree that when MORPH is trained on all pretraining datasets, then evaluating them on each of them individually can be considered a "zero-shot" setting. If all foundation models were pretrained on the same datasets, that would make the comparison much more useful, because then it would be possible to assess more clearly how well the transfer from pretraining datasets to finetuning datasets works.
>
> Author Response:
>
> Thank you for the clarification. In the paper, we used “zero-shot” to refer to evaluation on tasks drawn from the pretraining sets, **following the convention used in DPOT (Table 1)**. We agree that this usage can be confusing and is better described as “in-domain zero-shot.”
>
> Related to the question of pretraining sets: MORPH is designed to handle 1D/2D/3D datasets, as pointed by you was "is a very timely and relevant problem". Therefore, our pretraining and finetuning sets are bound by the restriction that we have to demonstrate the model capabilities of all data modalities. Exact alignment of pretraining sets becomes difficult in our case. Therefore, we perform fair comparisons on 2D datasets.
>
> **Concretely, we will**:
> - Along with existing nuances, we will explicitly highlight this aspect of fair comparison in Section 4.3.
> - Include a new table with “zero-shot transfer to fine-tuning sets,” i.e., evaluation on the fine-tuning datasets. (Table 4)
>
> Kindly let us know if any of your core concerns remain unaddressed. We will incorporate further clarifications and revisions with care.

---

> ### Author Response · Authors · 2025-11-21
> **Reply to reviewer comments: Weakness section (points 3,4,5)**
>
> Continue...
>
> 6. Reviewer comment:
> The main paper contains a lot of technical material that can be moved to the appendix in my opinion to make space for more relevant ablation studies and experiments. For example, details about the data stream and sharding setup as well as the compute infrastructure.
>
> Author Response:
> Thank you for this suggestion. Given the 9-page limit, some results were pushed to the appendix, but we agree that the main paper should prioritize ablations and core experiments. In the revision, we will streamline the main text and move implementation and infrastructure details to the appendix.
>
> Planned revisions:
> - Relocate Section 4.2 “Pretraining Setup” (data streaming, sharding, and compute details) to the appendix, keeping only a concise summary in the main text.
> - Surface additional experimental results from the appendix into the main paper to improve balance.
> - Add ablation studies isolating the roles of convolutions, cross-attention, and axial attention, reported as tables across three datasets (with metrics such as MSE), and include brief takeaways for each ablation.
>
> 7. Reviewer comment:
> While 3D datasets are considered, the resolutions are not very high. I expect the MORPH to scale well to high resolutions due to the axial attention, however I would appreciate a detailed model comparison regarding GFLOPs, throughput and VRAM requirements of the different networks.
>
> Author Response:
> - We agree with you that axial attention scales well with data resolutions. We have currently using datasets coming from three popular benchmarks: The Well (TW), PDEBench (PB), PDEGym (PG). We used 4, 3D datasets, from PDEBench and The Well in our pretraining and finetuning setup. 3D-MHD has the highest resolution 256x256x256 amongst of all of the available benchmarks, but we have used 64x64x64 version of the same dataset because $256^3$ MHD dataset is a upscaled version of $64^3$.
>
> - We have chosen 128x128x128 CFD dataset as our maximum resolution dataset, which gives us 4096 spatial tokens or patches with a patch-size of 8. These number of tokens are 4 to 8x higher than DPOT, MPP and Poseidon.
>
> - We will perform ablations and scaling studies on different datasets, comparing GFLOPs, parameters and accuracy as a part of the main paper. It will help resolve the concerns.
>
> 8. Reviewer comment:
> The architecture of MORPH seems to be very similar to MPP (axial attention over space+time). Can you clarify the differences and if they are very similar, do an ablation study for every modification? I would also appreciate an ablation study on different design choices of the network.
>
> Author Response:
> Thank you for this observation. We see several key differences between MPP and MORPH:
>
> a) Temporal windowing: MPP applies time-attention over a fixed 16-step window, which requires 16 input frames to begin rollout. MORPH supports axial attention over an arbitrary number of frames T. In our current foundation-model configuration we use T=1 for pretraining efficiency, but the design admits any T (Section 4.3).
>
> b) Spatial dimensionality: Based on the axial attention paper published in 2019 (AXIAL ATTENTION IN MULTIDIMENSIONAL TRANSFORMERS), it doesn't restrict the axial scaling to 3D datasets.   MPP is evaluated on 2D datasets and thus applies axial attention over (H, W, T). MORPH is designed for 1D-3D PDE datasets and extends axial attention to (T, D, H, W), without restricting axial attentions to 2D+time.
>
> c) Token regime: As you note, 3D PDE settings produce substantially more tokens. Axial attention enables efficient design in these higher-token regimes, which we target explicitly.
>
> Planned ablations:
> - We will provide ablations for each architectural choice, including axial attention vs. full attention vs. sparse attention, and report GFLOPs, parameter counts, and accuracy across three representative datasets.
>
> Please reach out to us if we are not able to address your comments and concerns properly. Also, thank you again for taking time to review our paper. Your expert comments have strengthened the paper.

---

> ### Author Response · Authors · 2025-11-21
> **Reply to reviewer comments: Weakness section (points 6,7,8)**
>
> Continue...
>
> 9. The results on autoregressive rollouts are not comprehensive; are the models stable for more than 10 autoregressive steps? The dynamics in Figures 4 to 7 do not show much change over time. Why didn't you show rollouts for 3D? Those are most interesting as they are most challenging.
>
> Authors' response:
>
> - We have shown autoregressive rollouts for 10 rollout steps on 2 separate datasets given only the first frame at t = 0. These two datasets are coming from PDEBench and PDEGym. We didn't quantify the variance over time for each dataset (because these are opensource datasets, widely used in the community).
>
> - Clarifying the comparison: prior works such as MPP/DPOT use multiple input frames (e.g., 16 and 10, respectively, as reported in their papers) before predicting future frames, whereas our experiments start from a single frame. We consider MORPH results in autoregressive rollouts as strong.
>
> - We would try to address this concern by presenting rollouts for more datasets.
>
> 10. It's not true that "this is the first study to apply LoRA to PDE foundation models", see [1]
>
> Authors' response:
> - Thank you for pointing this out. We missed this in the paper paper. The paper “PDE-Transformer” appeared on arXiv approximately couple of months before our submission, which likely explains why it was not included in our initial scan. We will consider it concurrent work, and we have highlighted this point in the manuscript.
>
> 11. The word "shape-agnostic" made me first believe the paper proposes a model that works on arbitrary geometries; the current title is overselling it to me as the model is still limited to regular grids.
>
> Authors' response:
>
> - Our use of “shape” is intended to denote “data shape.” In the abstract and introduction, we did not mention geometric aspects, nor did we inflate capabilities we have not investigated. We consistently refer to “shape” as “data shape.” However, we agree to the reviewer comment that title may put up a notion of geometry agnosticity in flow problems to some readers. To address this potential confusion, we would like to modify the paper’s title to “MORPH: PDE Foundation Models with Arbitrary Data Modality.” Please let us know what you think about this.
>
> Please reach out to us in case we are not able to address the comments. We will try to put our efforts to address it before the final submission.

---

> ### Author Response · Authors · 2025-11-26
> **Reply to reviewers' questions**
>
> Reviewer comments (questions):
> 1. How is the input history of 10 previous states in DPOT-FM handled in the zero-shot setting?
> 2. Have you considered any methods like Patch n' Pack https://arxiv.org/pdf/2307.06304 or others for training on different datasets and resolutions?
> 3. Have you looked at the performance of MORPH when changing the resolution of a specific pretraining dataset?
> 4. I did not understand if you are using ST-Bilinear or S-Linear & T-Slice for MORPH? Have you considered relative positional encodings, e.g. RoPE? Do you think this makes sense?
> 5. The information density of a patch with patch size 8 is much lower for 1D than for 3D data. Is this a problem?
> 6. Do you use any additional conditioning information like a class label for each dataset? Would this be helpful?
>
> Authors response:
>
> 1. DPOT-FM is pretrained with a 10-step temporal context. For standalone DPOT models (Table 2, top table), we have used the numbers presented in the DPOT paper. For our finetuning results (Table 2, bottom table), we keep the architecture unchanged and follow the official single-step adaptation: for each time step t, we feed ten copies of the current state $[x_{t},…,x_{t}]$ into DPOT and predict $x_{t+1}$. This is the same padding strategy used in POSEIDON for adapting DPOT to single-step prediction. We have described this in *Appendix. B.4.2*.
>
> 2. We do not use Patch n’ Pack or similar packing schemes in the current work. Heterogeneous datasets and resolutions are handled via the Unified Physics Tensor Format (UPTF-7) plus a fixed spatial patch size of 8 (can be changed and made adaptable) and axial attention, with a cap of 4096 max-patches across 1D–3D data. Patch-packing strategies are compatible with MORPH but would require reworking the 4D axial attention and dataloader. We see this as an orthogonal extension and plan to explore it, especially for even higher resolutions, like heliophysics and climate datasets.
>
> 3. We have not systematically investigated this aspect. In our current setup, all datasets are used at the native resolutions provided by PDEBench, The Well and PDEGym. We expect, these carefully curated benchmarks use the resolutions that respect the underlying complex physics, in particular by *resolving small-scale structures/high-frequency spectral components*. Particularly, the multi-physics (MHD, TGC) and turbulent CFD datasets represents complex multi-scale interactions, lowering the resolution would likely wash out relevant features. Exploring controlled changes of the pretraining resolution for a single dataset is an interesting direction, and we plan to study this in follow-up work.
>
> 4. In all reported experiments, MORPH-Ti/S/M use the S-Linear (linear interpolation in space) & T-Slice (no interpolation, since it's at a single time) *absolute positional encoding* with max autoregressive order 1, while MORPH-L uses the ST-Bilinear (bilinear interpolation in space and time) encoding with max order 16. These positional encodings are learnable. *We did not use relative positional encodings such as RoPE*. Our main requirement was a simple scheme that supports variable numbers of patches and time steps across heterogeneous 1D–3D grids. Incorporating RoPE-style relative encodings into our 4D axial attention blocks is technically straightforward and we agree it could be a promising direction, especially for longer temporal contexts.
>
> 5. We didn't see a patch-size of 8, presenting any problem for 1d datasets, however, we agree that it contains less information than an 8x8x8 3D patch. Empirically, *we do not observe degraded 1D performance on our 1D datasets*: 1D-CFD, 1D-DR, 1D-BE. Our  MORPH-SS and MORPH-FM match or improve on FNO and U-Net baselines.
>
> 6. We do not use any explicit conditioning such as dataset/PDE-family class labels. All datasets are mixed using balanced task sampling and normalized via dataset-specific ReVIN statistics, and the model only sees the fields in the unified UPTF-7 layout. Adding a dataset-ID or physics-ID token is straightforward and could help reduce interference between very dissimilar datasets. However, it would also encourage the model to rely on dataset-specific cues rather than learning a unified representation, which we expect *could hurt zero-shot transfer and fine-tuning to new PDE families*. We therefore chose a fully task-agnostic setup.
>
> Kindly let us know if any of your questions remain unaddressed. We will try to present further clarifications.

---

> ### Author Response · Authors · 2025-12-03
> **Reply to reviewer comments: Weakness section (point 2) New**
>
> **Reviewer comment**: While the paper is evaluated against other foundation models, the other foundation models are not pretrained with the same datasets. It would still be necessary to evaluate against other PDE foundation models when all have the same pretraining data.
>
> **Authors' response**: In response to the reviewers’ emphasis on this issue, we added a new experiment in Ablation Studies (Pretraining Corpus) (Sec. 4.4) specifically designed to address this comment.
>
> Write-up: To evaluate the effect of the pretraining set and to disentangle the influence of the dataset from that of the model architecture, we pretrain MORPH-FM-S*(27M) on the same datasets as Poseidon. We then finetune both MORPH and Poseidon on 128 trajectories and test on 240 trajectories for 200 epochs across three downstream tasks: CE-RM, NS-PwC, and NS-SL. The results reported in Table 9 (Appendix C.2) demonstrate that MORPH-FM-S*(27M), with only one-fifth the number of parameters, performs competitively with or even outperforms Poseidon. These findings indicate that the MORPH architecture yields performance gains that are largely independent of the choice of pretraining corpus. More details about this study are provided in Appendix C.2, and full autoregressive rollouts are shown in Figs. 10, 12 and 11 (Appendix C.3).

---

### Official Review · Reviewer_UJ5T · 2025-10-31

**Soundness:** 3
**Presentation:** 3
**Contribution:** 3
**Rating:** 6
**Confidence:** 3

**Summary:**

The paper introduces MORPH - a PDE foundation model that handles data of varying dimension at different resolutions and with mixed scaler and vector components.
The paper presents a flexible foundation model that can learn from diverse forms of partially observed scientific data.
The model supports shape agnostic with physics-aware channel handling of PDE datasets. This is done by using a) component-wise convolutions for local interactions b) interfield cross attention for modeling information across physical fields
c) axial attention for factorizing full spatiotemporal self attention along spatial and temporal axes, reducing compute while retaining expressivity. The paper defines a tensor format for minibatches that generalizes across PDE datasets while preserving semantics.
pretraining is done on six heterogeneous datasets. full finetuning and parameter efficient fine tuning done with LoRA on seven additional datasets. The largest model has 480M parameters.

**Strengths:**

PDE foundation models are an important development and this method attempts to overcome limitations of prior methods giving equal weight to heterogeneous data sources.

The method proposes a unified tensor format to handle sources of varying shapes and scalar and vector fields.

Model features are chosen for efficient and expressive processing of PDE fields where convolutional operations capture locality, fieldwise attention promotes interaction between fields and Factorized spatiotemporal attention in axial attentions makes for efficient processing.

Multiple models are trained with varying parameters counts and compared including with full and low rank fine tuning.

Experiments are performed on a number of datasets.

The paper is mostly clear and well written.

**Weaknesses:**

The unified tensor format handles fields that have varying the number of components by broadcasting to match vector components which appears to be wasteful.

The details of the operations of interfield attention and axial attention are vague from their description in the main paper. Some clarity would be useful.

**Questions:**

The paper mentions partial observability in the challenges with PDE modeling. How does MORPH handle this?

What mixed scalar and vector fields are used in the experiments? Do you assume at most 3d vector fields? What are the dimensions of the vector components?

line 217 how are the different data processing components combined into the transformer architecture. Is the flow sequential as convolution-> field wise -> axial -> decoding or is it more complex?

line 99. What is meant by attention chaos?

---

> ### Author Response · Authors · 2025-11-20
> **What mixed scalar and vector fields are used in the experiments? Do you assume at most 3d vector fields? What are the dimensions of the vector components?**
>
> Thank you for the thoughtful comments. We appreciate your time reviewing our article. We want to address all of your comments (including comments from other reviewers) with best possible intensions, clarity and efforts.
>
> Reviewer’s question:
> "What mixed scalar and vector fields are used in the experiments? Do you assume at most 3D vector fields? What are the dimensions of the vector components?"
>
> - Across different datasets, we have multiple scalars and vector fields. For ex: MHD has 2 vector fields (velocity: Vx,Vy,Vz and magnetic fields: Bx, By, Bz) and 2 scalar fields (pressure and density fields).  Prior work  flattens fields and components into channels for a vision transformer and model treats them equally. We instead use the UPTF-7 layout (B, T, F, C, D, H, W), which separates fields (F) from components (C). Components pass through a convolutional block that produces eight feature maps, while fields are fused via cross-attention ("field-fusion").
>
> - Why separate fields and components? Cross-attention fuses each field’s components into a single representation, which is computationally efficient for 3D data. If every channel were treated as a separate field (e.g., 10 fields in MHD), the required fusion would over-compress information and degrade predictions. Convolution over components captures locality and inductive bias while introducing fewer parameters and FLOPs than cross-attention.
>
> - On lifting scalars to vectors. In this submission, we temporarily lift scalar fields to vector fields to simplify batching during streaming. This adds a small batch-level overhead. Our framework can support mixed scalar and vector inputs without lifting, but that is not implemented in the current submission and remains a limitation.
>
> Revisions in the paper:
> 1) Table 1 and 2 to explicitly indicate which fields are scalar versus vector for each dataset.
> 2) Clearly describe the scalar-to-vector lifting in the Limitations section.
>
> Please let us know if this addresses your question. We are also making additional revisions throughout the paper to respond to your other comments.

---

> ### Author Response · Authors · 2025-11-20
> **Reply to reviewer comment: line 217 how are the different data processing components combined into the transformer architecture. Is the flow sequential as convolution-> field wise -> axial -> decoding or is it more complex?**
>
> Reply to reviewer comment: line 217 how are the different data processing components combined into the transformer architecture. Is the flow sequential as convolution-> field wise -> axial -> decoding or is it more complex?
>
> You picked it right. The flow of data processing through different operators of our network is sequential. More detailed description:
>
> - The UPTF-7 tensor arrives at the convolution operator where components ("C") transforms into feature maps. The UPTF-7 transforms into (B,T,F,filters,D,H,W).
>
> - Field-wise patching is performed to get the features = patch x filters and tokens as (D,H,W)/patch_sizes for each field.
>
> - Field-wise embedding: A linear layer maps features to transformer model dimensions (d_model: 256 for Ti, 512 for S, 768 for M and 1024 for L model).
>
> - Cross-attention: Fuses multiple fields to 1 field. By the end of this layer we get (batch, t, tokens, features).
>
> - Transformer layers: The transformer produces (batch, t, tokens, features) where attention is axially performed on t, spatially decomposed tokens (attentions of d tokens, h tokens, w tokens).
>
> - Decoder: Get's batch UPTF-7 format.
>
> Please feel free to reach out if something is missing and not well addressed in your reply.

---

> ### Author Response · Authors · 2025-11-20
> **Reply to (1) line 99. What is meant by attention chaos? (2) The paper mentions partial observability in the challenges with PDE modeling. How does MORPH handle this?**
>
> 1) Reviewer comment: Comment (line 99): “What is meant by attention chaos?”
>
> Authors response:
>
> -We originally used “attention chaos” as a loose analogy to “polynomial chaos expansion” to describe attention computed over multiple axes (fields and spatial dimensions: F, D, H, W). Because this term is uncommon and insufficiently justified in the paper, we will replace it with a clearer phrase e.g., “multi-axis attention over (F, D, H, W)” and add a one-sentence definition for readability. Thank you for pointing this out.
>
> 2) Reviewer comment: Partial observability: How does MORPH handle this?
>
> Authors response:
>
> - By “partial observability,” we mean that available data provide only limited, heterogeneous views of the physical state: datasets may be 1D/2D/3D, often from simulations with modeling assumptions and numerical errors, while real experiments are frequently 1D.
>
> - MORPH addresses this by (i) using a unified representation (UPTF-7: B, T, F, C, D, H, W) that accommodates 1D/2D/3D data and mixed scalar/vector fields, and (ii) applying field-aware fusion (cross-attention across fields) together with local convolution over components to preserve inductive bias. Pretraining across heterogeneous datasets followed by fine-tuning enables the model to learn from limited data and generalize to unseen fine-tuning sets.
>
> Steps taken to improve the paper
> 1) Improve the readability and remove the loose analogy.

---

> ### Author Response · Authors · 2025-11-21
> **Reply to reviewer comments: Weakness section**
>
> 1. Reviewer comment: The unified tensor format handles fields that have varying the number of components by broadcasting to match vector components which appears to be wasteful.
>
> Authors's reply:
>
> - We have replied to this comment, but we like to mentioned it in this context as well. We lift scalar fields to vector fields with a physics-based assumption that every scalar fields is a particular case of vector fields. In this submission, we temporarily lift scalar fields to vector fields to simplify batching during streaming. This adds a small batch-level overhead.
>
> - Our framework can support mixed scalar and vector inputs without lifting, but that is not implemented in the current submission and remains a limitation. This limitation, we will outline in the limitations section of the paper.
>
> 2. Reviewer Comment: The details of the operations of interfield attention and axial attention are vague from their description in the main paper. Some clarity would be useful.
>
> Authors' response:
>
> - Thank you for pointing it out. Currently, we are performing ablation studies on interfield attentions and axial attentions on 3 different datasets. It will help with the readability and may help in addressing reviewer concerns.
>
> - In response to the other reviewers’ concerns, we have reorganized the manuscript with clearer tables and a more structured presentation of technical details.
>
> Please let us know if any of our replies are unclear and your concerns remain unaddressed. We will continue refining the manuscript up to the final submission to ensure we clearly communicate the novelty and contributions of MORPH to the research community.

---

### Author Response · Authors · 2025-12-03
**Minor updates/changes in the paper**

### ----- Minor updates in response to reviewer comments -----
#### (*We list only the reviewer comments that resulted in *minor* changes to the paper.)

#### 1. Reviewer UJ5T comment (weakness point 1, 2; Question 2)
#### 1.1 What mixed scalar and vector fields are used in the experiments? Do you assume at most 3D vector fields? What re the dimensions of the vector components?

#### 1.2 The unified tensor format handles fields that have varying the number of components by broadcasting to match vector components which appears to be wasteful.

#### **Updates in the paper**:
#### *What? and Where?*
#### - To improve readability, we split the original Table 1 into two separate tables (Tables 1 and 2). The new tables explicitly define the pretraining and fine-tuning sets together with their fields and component structure.
#### - We added a clarification on scalar-to-vector lifting in UPTF-7 in the Limitations section (at the end of the paper).

#### 2. Reviewer UJ5T comment (Question - 4): What is meant by attention chaos?

#### **Updates in the paper**:
#### *What? and Where?* We removed this loose “attention chaos” analogy from the paper and updated the contributions paragraph on Page 2 accordingly.

#### 3. Reviewer hKzL comment (Weakness point 1): FNO/U-Net benchmarks speculation

#### **Updates in the paper**:
#### *What? and Where?* We added the clarifying sentence “The results for baselines and SOTA models are inherited from the respective benchmark sources (Takamoto et al., 2022; Ohana et al., 2024) and articles (McCabe et al., 2024; Hao et al., 2024)” in Sec. 4.2 to make clear that we report benchmarked values from the original sources rather than reimplementing all baselines ourselves.

#### 4. Reviewers' hKzL (Weakness point 1) and 53go comment (Weakness point 1): Reconfiguration of the paper

#### **Updates in the paper**:
#### *What? and Where?* We restructured the paper to improve the flow between main results and appendices. We moved several key results from Appendix C into the main paper. We shifted Sec. 3.2 and Sec. 4.2 into Appendices A.4 and B.2, respectively, to reduce redundancy and keep the main narrative focused.

#### 5. Reviewers' hKzL (Weakness point 2) and 53go comment (Weakness point 2): MORPH pretraining corpus is different than DPOT and Poseidon.

#### **Updates in the paper**:
#### *What? and Where?* We performed a new study on this in the ablation section, as described above in the “Pretraining corpus mismatch” item. In addition, we added a new paragraph titled Comparison Caveats in Sec. 4.1 that explicitly discusses the challenges of making perfectly matched comparisons with existing PDE foundation models (DPOT, Poseidon).

#### 6. Reviewer hKzL comment (Weakness point 8): The phrase “shape-agnostic” refers to regular grids and geometry-agnostic aspects.

#### **Updates in the paper**:
#### *What? and Where?* We revised the title of the paper to "MORPH: PDE Foundation Models with Arbitrary Data Modality" to more accurately describe the scope of the method and avoid confusing “shape-agnostic” with geometry-agnosticity in fluid problems.

#### 7. Reviewer hKzL comment (Weakness point 7): LoRA is used in earlier work.
#### *What? and Where?* Sec. 4.2 now includes the sentence “To our knowledge, this is one of the first two concurrent studies to apply LoRA to PDE foundation models (see also Holzschuh et al. (2025)).” to acknowledge concurrent work and to more precisely position our contribution.

---

### Author Response · Authors · 2025-12-03
**Major updates/changes in the paper (part-2)**

Continue...

#### 3. **Reviewers' comment: Scaling studies**

#### 3.1 Reviewer hKzL: Weakness point - 1.

#### 3.2 Reviewer 53go: Weakness point - 1.

#### **Updates in the paper**:

#### *Where?* Write-up: Sec. 4.3 Scaling studies, Appendix. C.1; Results: Fig. 4, Fig. 5, Table 8.

#### *What?* We now present two new pretraining scaling studies in addition to the two existing fine-tuning scaling studies. Specifically, we study (i) scaling with respect to the size of the pretraining dataset and (ii) scaling with respect to the pretraining model size. The fine-tuning scaling studies, which were previously in the appendix, have been moved to the main paper.

#### *Take-away*
#### - Fine-tuning errors consistently decrease as the fine-tuning dataset size increases.
#### - MORPH shows better scaling efficiency than existing PDE foundation models over comparable data and compute budgets.
#### - Pretraining losses decrease steadily as the pretraining dataset size increases from 25% to 100% of the corpus.
#### - Under a fixed training budget of 10 epochs and a shared training protocol (same learning rate, batch size, and weight decay), we do not observe clear gains when scaling model size from Ti to L, suggesting that mild model-specific hyperparameter tuning is likely needed to fully realize the benefits of larger MORPH models. A more exhaustive hyperparameter search is left for future work.

#### 4. **Reviewers' comment: Transfer across modalities - can MORPH really profit from other-dimensional data?**

#### 4.1  Reviewer 53go: Weakness point - 3.

#### **Updates in the paper**:

#### *Where?* Write-up: *Sec. 3.3 Transfer across data modalities*; Results: Fig. 2.

#### *What?* We added a new study in which we pretrain MORPH exclusively on the 2D incompressible Navier–Stokes dataset (CFD2D-IC). We then perform zero-shot inference (no fine-tuning, all weights frozen) on six heterogeneous, fluid-based target datasets spanning different modalities: CFD1D, CFD2D, CFD3D, CFD3D-TURB, MHD3D, and TGC3D. To quantify cross-modality transfer, we define the Gap Closure Ratio, which measures how much of the performance gap between a naive, training-free baseline and a fully trained-from-scratch model is closed by the zero-shot pretrained model.

#### *Take-away*  Fig. 2 shows that 2D pretraining learns reusable operators that transfer to 1D, 2D, and 3D data modalities, providing empirical support for MORPH’s modality-agnostic design.

#### 5. **Reviewers' comment: Rollouts**

#### 5.1. Reviewer hKzL: Weakness point - 6.

#### **Updates in the paper**:

#### - *Where?* Write-up: Sec. 4.2 Main Results (Autoregressive rollouts), Appendix C.3; Results: Fig. 3, Figs. 6-12.

#### - *What?* We moved Fig. 3 from the appendix into the main paper. In addition, we added new rollout results on CE-RM, CE-SL, and CE-PwC (Figs. 10–12), where we present full autoregressive rollouts for all fields.

#### - *Take-away* In both the pretrained and fine-tuned stages, MORPH produces stable long-horizon autoregressive rollouts, even at smaller model sizes, directly addressing concerns about rollout quality and stability.

#### 6. **Reviewers' comment: Zero-shot on fine-tuning sets**

#### 6.1. Reviewer hKzL: Weakness point - 2.

#### **Updates in the paper**:

#### - *Where?* Sec. 4.1, Table 4.

#### - *What?* We added new zero-shot evaluation results on the fine-tuning datasets, in addition to the existing zero-shot results on the pretraining datasets. These results are summarized in Table 4.

#### - *Take-away* MORPH-FM-L achieves the best performance on 11 out of 12 datasets compared to smaller MORPH variants. This indicates favorable scaling behavior of MORPH under zero-shot adaptation and strengthens our claim that larger MORPH models generalize more robustly across diverse PDE benchmarks.

---

### Author Response · Authors · 2025-12-03
**Major updates/changes in the paper (part-1)**

### Summary of changes

#### Our key contribution is MORPH, a PDE foundation model that can jointly handle 1D, 2D, and 3D datasets within a single unified framework - a novel direction compared to prior PDE foundation models that typically specialize to a fixed dimensionality. In response to the reviews, we substantially strengthened the empirical support for this claim by adding a matched-corpus comparison to Poseidon, extensive architectural ablations, new transfer and pretraining/fine-tuning scaling studies, and by moving key experimental results into the main paper. The experiments show that a 27M-parameter MORPH model, pretrained on exactly the same corpus as Poseidon, can match or surpass a 158M-parameter Poseidon model on most PDEGym benchmarks. MORPH exhibits favorable scaling and zero-shot transfer behavior across 11/12 datasets. We further added a dedicated cross-modality transfer study demonstrating that a 2D-pretrained MORPH learns reusable operators that generalize across 1D/2D/3D problems and produce stable long-horizon dynamics even at smaller model sizes. We have expanded rollout visualizations. Finally, we systematically addressed detailed concerns about mixed scalar–vector fields, scalar-to-vector lifting, benchmarking and comparison caveats, terminology, and concurrent LoRA work, so that the paper is both novel in scope and more transparent, fair, and rigorous in its empirical methodology.

### ----- Major updates in response to reviewer comments -----
#### (*We list only the reviewer comments that resulted in *major* changes to the paper.)

#### 1. **Reviewers' comment: Pretraining corpus mismatch and fairness of comparison**

#### 1.1 - Reviewer hKzL: Weakness point 2.

#### 1.2 - Reviewer 53go: Weakness point 2.

#### **Updates in the paper**:

#### *Where?* Write-up: Sec. 4.4 Ablation Studies (Pretraining corpus), Appendix C.2; Table 9, Figs. 10-12.

#### *What?* We added a new ablation study where we pretrain MORPH on exactly the same pretraining corpus as Poseidon, i.e., NS-Sines, NS-Gaussians, CE-RP, CE-CRP, CE-KH, and CE-Gauss. We then compare a 27M-parameter MORPH variant (MORPH-FM-S*) against Poseidon-FM-B (158M) on three PDEGym fine-tuning tasks: CE-RM, NS-PwC, and NS-SL, under a shared fine-tuning protocol. In this matched-corpus setting, MORPH-FM-S* clearly outperforms Poseidon-FM-B on CE-RM and NS-SL, while Poseidon performs better on NS-PwC. Full rollout results for these experiments are reported in Table 9 and Figs. 10–12.

#### *Take-away*
#### - This new experiment directly addresses the concern about pretraining corpus mismatch and shows that MORPH’s architectural choices yield strong performance even when the pretraining data are held fixed.
#### - With the same pretraining corpus as Poseidon and ∼5× fewer parameters, MORPH produces stable autoregressive rollouts and strong downstream performance in data- and compute-scarce fine-tuning scenarios.
#### - These results support using MORPH as a lightweight PDE foundation model that can match or surpass a substantially larger Poseidon-FM-B model on two out of three PDEGym benchmarks.

#### 2. **Reviewers' comment: Ablation studies**

#### 2.1 - Reviewer hKzL: Weakness point- 3, 5.

#### 2.2 - Reviewer 53go: Weakness point- 1.

#### **Updates in the paper**:

#### *Where?* Write-up: Sec. 4.4 Ablation Studies (Model architecture), Appendix C.2.; Results: Table 6

#### *What?* We added three ablation studies on the model architecture, varying (i) convolutional layers, (ii) cross-attention for field fusion, and (iii) axial attention. These ablations are conducted on five different datasets, and we compare validation loss, GFLOPs, and parameter counts across these design choices.

#### *Take-away*
#### - Adding convolutions improves the loss with only a modest increase in GFLOPs.
#### - Cross-attention–based field fusion consistently improves accuracy with negligible parameter overhead and only a moderate increase in FLOPs compared to the “concat + dense” baseline.
#### - Axial attention offers the best accuracy–compute trade-off among full and sparse attention variants, and its advantage grows with sequence length.

---

### Note · Authors · 2026-03-24

I have read and agree with the venue's withdrawal policy on behalf of myself and my co-authors.

---

### Meta-Review · Area_Chair_CKYo · 2025-12-19

**Summary:**

This paper proposes a foundation model for PDEs, with the key advantage of not limiting to a fixed dimensionality, supporting up to 3D data.

One key concern was the baseline choices and evaluation. The authors rebutted that they have used some standard choices, but nevertheless did not extend the results. I understand the point that different models handle different data types, but instead of a “head-to-head” comparison, I would agree with hKzL. The authors should try to compare with the “best-in-class” models and show that their approach is better. The comparison with prior models was also raised by 53go. This was the central criticism of the work, which was not addressed by the rebuttal. Therefore, I recommend rejection.

**Reviewer Concerns:**

The main recurrent reviewer's concern is on the evaluation, in particular the baselines choice and how they were trained. I think these points are valid and the authros did not address them.

**Reviewer Scores:**

I don't think the reviewers will have changed their scores based on the rebuttal as the main concerns were not addressed.

---

### Decision · Program_Chairs · 2026-01-26

Reject